# Vision Foundation Models as Effective Visual Tokenizers for Autoregressive Generation

**Anlin Zheng**[1]    **Xin Wen**[1]    **Xuanyang Zhang**[2*]    **Chuofan Ma**[1]
**Tiancai Wang**[3]    **Gang Yu**[2]    **Xiangyu Zhang**[2,4]    **Xiaojuan Qi**[1*†]

[1]The University of Hong Kong    [2]StepFun    [3]Dexmal    [4]MEGVII Technology

## Abstract

In this work, we present a novel direction to build an image tokenizer directly on top of a frozen vision foundation model, which is a largely underexplored area. Specifically, we employ a frozen vision foundation model as the encoder of our tokenizer. To enhance its effectiveness, we introduce two key components: (1) a region-adaptive quantization framework that reduces redundancy in the pre-trained features on regular 2D grids, and (2) a semantic reconstruction objective that aligns the tokenizer's outputs with the foundation model's representations to preserve semantic fidelity. Based on these designs, our proposed image tokenizer, **VFMTok**, achieves substantial improvements in image reconstruction and generation quality, while also enhancing token efficiency. It further boosts autoregressive (AR) generation—achieving a gFID of **1.36** on ImageNet benchmarks, while accelerating model convergence by **three times**, and enabling high-fidelity class-conditional synthesis without the need for classifier-free guidance (CFG). The code is available at https://github.com/CVMI-Lab/VFMTok.

## 1 Introduction

GPT's success in language generation has spurred interest in autoregressive (AR) image generation [42, 43, 49], which relies on visual tokenizers like VQGAN [13, 38, 42, 47, 51] to map images into compact, discrete latent spaces. However, these tokenizers, typically trained from scratch and optimized for reconstruction, often yield latent spaces rich in low-level details but poor in high-level semantics and laden with redundancy. Such flawed latent spaces not only prolong AR model training (Fig. 1) but also necessitate techniques like classifier-free guidance (CFG) for high-fidelity, class-conditional image generation, which in turn increases inference time.

In parallel, within the field of computer vision, pre-trained vision foundation models such as DINOv2 and CLIP [9, 33, 36, 45, 57] have demonstrated strong capabilities in extracting semantically rich and generalizable visual features. Early explorations in diffusion-based image generation—*e.g.*, REPA [56]—suggest that the semantic representations learned by these models can facilitate the training of generative models. This leads to a natural and compelling question: *Can the latent features from vision foundation models, originally designed for visual understanding, also serve as robust and structured representations for image reconstruction and generation?*

Recent studies [61, 63] have started exploring this direction by leveraging features from vision foundation models to initialize quantizer codebooks [61, 63], augment VQGAN architectures with additional branches [35], or distill these features to guide latent space learning [40]. Although these approaches show promise, they typically treat foundation model features as auxiliary components rather than fully capitalizing on their potential as generative priors. As a result, these methods often suffer from inefficiencies and fail to fully utilize the rich semantic information embedded in foundation model features, leaving their generative capabilities largely underexplored.

---

[†] Corresponding author: xjqi@eee.hku.hk. [*] Project lead.

39th Conference on Neural Information Processing Systems (NeurIPS 2025).

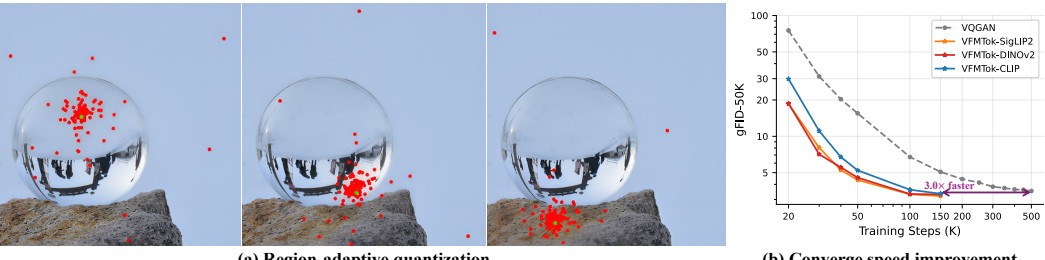

| (a) Region-adaptive quantization | (b) Converge speed improvement |

Figure 1: VFMTok introduces novel features, including: a).**region-adaptive quantization**— where it adaptively samples regions of similar patterns and extracts their VFM features for quantization; b).**convergence speed improvement** compared with vanilla VQGAN [42] for AR image synthesis.

**Can VFMs be effective tokenizers?** To address this, we initialized the encoder of a VQGAN with different frozen pre-trained foundation models to reconstruct images. Once trained, the tokenizer is integrated on top of an AR model for image synthesis (implementation details depicted in Sec. 3.2) As shown in Tab. 1 (middle rows), our results demonstrate that **the semantically rich features from these foundation models not only support effective image reconstruction but also achieve generative performance comparable to—or even surpassing—that of a fully trained VQGAN encoder optimized for both reconstruction and generation**. These findings highlight the strong potential of pre-trained vision foundation models to serve as efficient and effective tokenizers for image generation tasks, eliminating the need for extensive encoder training while improving qualities.

**Can we improve token efficiency for VFMs?** Building on this pilot study, we are further motivated by the observation that natural images often consist of irregular regions that exhibit recurring visual patterns. For example, as illustrated in Fig. 1(a), the upper portion of the crystal ball exhibits consistent patterns such as texture and transparency; similarly, the moss in the stone possesses similar textural structure. When such images are represented using a regular 2D feature grid extracted from foundation models, this structure-agnostic representation may introduce significant redundancy. Exploiting redundancy within semantically coherent regions offers a promising direction for improving tokenization efficiency. Motivated by this insight, we propose a region-adaptive strategy to refine the latent space that aims to enhance both image reconstruction and generation quality while significantly improving token representation efficiency.

**Our solution and results.** Guided by the preceding experimental analysis and insights, we introduce VFMTok, an image tokenizer that leverages a frozen pre-trained vision foundation model for adaptive region-level tokenization. VFMTok is designed to achieve high reconstruction and generation quality with improved token efficiency. Specifically, VFMTok employs a frozen pre-trained VFM as an encoder to extract multi-level semantic features. A set of learnable anchor queries performs region-level sampling on these features via de-

Table 1: Pilot study of image reconstruction and generation on ImageNet [10]. Relative wall-clock inference time for the tokenizer (compared to VFMTok) is reported. L.P. denotes linear probing results on the ImageNet validation set, used to estimate the semantic quality of latent tokens.

| Setup | Image Recon. | | | AR Generation | | | L.P. |
|---|---|---|---|---|---|---|---|
| | #Tok. | rFID↓ | rIS↑ | gFID↓ | gIS↑ | Time | (%) |
| VQGAN [42] | 576 | 0.95 | 197.3 | 3.71 | 228.3 | 4.3 | 23.1 |
| VQGAN (CLIP) | | 1.47 | 182.0 | 3.45 | 221.2 | 4.0 | 59.5 |
| VQGAN (SigLIP2) | 576 | 0.96 | 198.4 | 3.39 | 267.8 | 4.0 | 55.5 |
| VQGAN (DINOv2) | | 0.99 | 206.3 | 3.34 | 268.6 | 4.0 | 56.4 |
| VFMTok (CLIP) | | 0.99 | 200.1 | 3.40 | 274.7 | 1.0 | 63.9 |
| VFMTok (SigLIP2) | 256 | 0.94 | **218.7** | **3.01** | **280.8** | 1.0 | **78.5** |
| VFMTok (DINOv2) | | **0.89** | 215.4 | 3.08 | 274.2 | 1.0 | 69.4 |

formable attention [62], producing region-adaptive tokens that are subsequently quantized into discrete tokens representing the image's latent representation. These contextual tokens are then processed by a lightweight Vision Transformer [12](ViT) in a BERT-style framework [11, 15] with two primary reconstruction objectives. First, the original image pixels are reconstructed after dequantization using a VQGAN [42] decoder. Then, the model reconstructs the features from the frozen foundation model itself, allowing VFMTok to retain the semantic richness and discriminative power of the original representations. Once trained, VFMTok enables standard autoregressive Transformers (*e.g.*, Llama [44]) to generate contextual token sequences, which are decoded back into images via the VQGAN decoder, facilitating high-quality image synthesis with compact and semantically mean-

ingful representations. As shown in Tab. 1 (bottom rows), VFMTok achieves superior reconstruction and generation performance while using fewer than half the original number of tokens (256 *vs.* 576).

Extensive experiments validate that VFMTok, by combining the representational power of visual foundation models with a novel region-adaptive tokenization strategy based on irregular sampling and learnable anchor queries, enables both high-quality and efficient image reconstruction and autoregressive (AR) generation. First, VFMTok achieves superior reconstruction quality and captures richer semantics using significantly fewer tokens compared to prior methods (*e.g.*, 256 *vs.* 576 in [42]), resulting in a structured, semantic-aware, and compact latent space. As shown in Tab. 1, VFMTok, with only 256 tokens, outperforms other tokenizers using the same VFM encoder by delivering superior reconstruction quality and stronger semantic representation (as indicated by linear probing). Second, the high-quality latent space produced by VFMTok facilitates effective AR training using a simple LLaMA-based model, leading to faster convergence (see Fig. 1(b)) and improved generation performance. Notably, the 1.4B AR model surpasses the performance of LlamaGen-3B despite having fewer parameters and requiring fewer training iterations. The 1.5B advanced AR model achieves a new state-of-the-art with a gFID of **1.36** on ImageNet [10] $256 \times 256$, outperforming widely-used diffusion models. Third, due to the compact token space and the reduced number of tokens, VFMTok significantly improves the inference speed of AR models (see Tab. 1). Moreover, the rich semantic content embedded in the latent tokens enables effective class-conditional image synthesis with high fidelity—without the need for classifier-free guidance—further reducing inference time.

Our contributions can be summarized as follows:

- We demonstrate that frozen vision foundation models—ranging from self-supervised to language-supervised—are effective for image reconstruction and generation. Leveraging their semantic richness enhances the tokenizer and enables AR generation models to converge faster and perform high-fidelity, CFG-free image synthesis, without bells and whistles.
- We propose a region-adaptive tokenization framework that effectively leverages inherent redundancies in image regions to achieve compact tokenization. This approach reduces the number of visual tokens while enhancing performance, enabling efficient AR generation without sacrificing quality.
- Extensive experiments validate the effectiveness of our approach in both image reconstruction and AR generation, establishing pre-trained vision foundation models as powerful tokenizers for high-quality and efficient image generation.

## 2  Related Work

**Image Tokenization using Autoencoders.** Pixel-space images are highly redundant. Autoencoder-based tokenizers [29, 42, 43, 53] create compact latent tokens to reduce redundancy. VQVAEs [21, 38, 47] and their derivatives evolved using adversarial losses [13], Transformers [51], multistage quantization [24, 59], lookup-free methods [29, 31], and codebook initialization from pre-trained features [61, 63]). These 2D tokenizers map features to a static 2D grid, which limits redundancy exploration. Recent 1D tokenizers [1, 32, 48, 55] offer superior compression, reconstruction, and redundancy removal, but often require complex and lengthy training. For example, TiTok [55] requires a two-stage process (warming up and fine-tuning) for 200 epochs. Our VFMTok adopts a novel region-adaptive tokenization framework to reduce redundancy. With a simpler training strategy for only 50 epochs, VFMTok exhibits discriminative semantics and excellent generation results.

**Vision Foundation Models.** Vision Foundation Models (VFMs) [3, 6, 14, 16, 17, 20, 33, 36, 45, 57] aim to learn general, transferable visual representations from large-scale, diverse data. The training of these versatile models has shifted from early supervised approaches to more scalable self-supervised learning [3, 6, 9, 11, 14–16, 33], which leverages inherent data structures. More recently, language-supervised pre-training [20, 45, 57] on vast image-text pairs has enabled VFMs to learn rich, semantically grounded representations. Pre-trained VFMs serve as powerful backbones for a wide array of downstream tasks. In this work, we utilize pre-trained VFMs directly as image tokenizers for AR image generation, surpassing other methods [61, 63] with superior performance. Furthermore, using VFMs as tokenizers enables the removal of classifier-free guidance.

**Autoregressive Image Generation.** GPT-style Transformers [5, 24, 37, 42, 43, 46] have spurred interest in autoregressive (AR) image generation, which predicts visual token sequences. While early AR models operated in pixel space [5, 46], current methods [24, 42, 43, 51] generate discrete latent tokens via next-token prediction, then decode them to pixels using a tokenizer's decoder [13, 38, 47, 51]. To improve the generation quality, recent works [25, 43, 49] add bidirectional attention (*e.g.*, VAR's

next-scale prediction [43], MAR's BERT-style framework [25], Show-o's hybrid attention [49]). These innovations, however, complicate designing universal, multi-modal Transformers adhering to next-token prediction. Instead, our VFMTok enables standard AR transformers to generate contextual token sequences for subsequent decoding, eliminating complex structural modifications.

## 3  Method

In this section, we first provide preliminary background on quantized image tokenizers. We then present our pilot studies exploring the use of vision foundation models for tokenization. Finally, we introduce VFMTok, a novel tokenizer built upon frozen vision foundation models, incorporating region-adaptive strategies to enhance both the efficiency and effectiveness of the tokenization process.

### 3.1  Preliminary

**Quantized Image Tokenizer.**    To apply autoregressive modeling to visual generation, existing methods [42, 43, 51, 52] necessitate an image tokenizer to map a 2D image into discrete token sequences for AR generation. To achieve this, quantized autoencoders, such as VQVAEs [13, 38, 42, 43, 47, 51, 61], are widely used. Typically, an image tokenizer consists of an encoder $\mathcal{E}(\cdot)$, a quantizer $\mathcal{VQ}(\cdot)$, and a decoder $\mathcal{D}(\cdot)$. Given an input image $\mathrm{I} \in \mathbb{R}^{H \times W \times 3}$, the encoder $\mathcal{E}(\cdot)$ first convert an image into patch embeddings $Z_{2D} \in \mathbb{R}^{\frac{H}{f} \times \frac{W}{f} \times \mathrm{D}}$ with spatial down-sampling factor $f$. Then, $Z_{2D}$ is fed into the quantizer $\mathcal{VQ}(\cdot)$ that includes a learnable codebook $\mathbb{C} \in \mathbb{R}^{N \times D}$ with $N$ vectors. Each feature vector $z_i \in \mathbb{R}^D$ is mapped into its nearest vector $c_i \in \mathbb{R}^D$ in the codebook $\mathbb{C}$.

$$Z_{2D} = \mathcal{E}(\mathrm{I}),$$
$$\mathcal{VQ}(z_i) = c_i, \quad \text{where} \quad i = \underset{j \in \{1,2,...,N\}}{\arg\min} \|z_i - c_j\|_2, \tag{1}$$

where $H$ and $W$ denote the input image's height and width, respectively. $D$ depicts the latent feature dimension. Once discrete tokens are acquired, they can be de-quantized into corresponding code and converted back to image pixels by the decoder $\mathcal{D}(\cdot)$, as depicted in Eq. (2).

$$\hat{\mathrm{I}} = \mathcal{D}(\mathcal{VQ}(Z_{2D})). \tag{2}$$

To optimize the codebook, the training objective is $\mathcal{L}_{\mathbf{vq}} = \sum \|\mathbf{sg}(z_i) - c_i\|_2^2 + \beta \cdot \|\mathbf{sg}(c_i) - z_i\|_2^2$, where $\mathbf{sg}(\cdot)$ is a stop-gradient function [2, 47]. The second term is a commitment loss weighted by $\beta$ to align extracted features with codebook vectors. For image reconstruction, the loss function is $\mathcal{L}_{AE} = \mathcal{L}_2(\mathrm{I}, \hat{\mathrm{I}}) + \mathcal{L}_P(\mathrm{I}, \hat{\mathrm{I}}) + \lambda_G \cdot \mathcal{L}_G(\hat{\mathrm{I}})$, where $\mathcal{L}_2$ is a pixel-wise reconstruction loss, $\mathcal{L}_P$ is perceptual loss from LPIPS [58], and $\mathcal{L}_G$ is adversarial loss from PatchGAN [19] with weight $\lambda_G$.

### 3.2  Pilot Study: Pre-trained Vision Foundation Models as Tokenizers for AR Generation

To assess whether a pre-trained VFM can serve as a tokenizer for image reconstruction and benefit image generation, we performed a pilot study. In our setup, we extract the final 2D grid features from images of size $336 \times 336$ using a frozen VFM, such as DINOv2, CLIP, and SigLIP2. These features, after quantization, are fed into a VQGAN [42] decoder for image reconstruction. Once trained, the tokenizer is integrated on top of a Llama-based AR model for image synthesis. Additionally, the training duration for VQGANs and AR models is 50 and 100 epochs, respectively.

As Tab. 1 illustrates, directly using features from pre-trained VFMs yields decent image reconstruction and generation performance compared to vanilla VQGANs. Notably, these VFM-based tokenizers consistently exhibit stronger semantic representation capabilities (as indicated by the linear probing experiment in Tab. 1). For instance, VQGAN (SigLIP2) achieves reconstruction performance on par with vanilla VQGAN, while exhibiting better semantic representation and superior generation quality. Nevertheless, variations in image reconstruction and generation quality arise when different VFMs are used to initialize the tokenizer's encoder. Specifically, VQGAN (DINOv2) and VQGAN (SigLIP2) demonstrate similar reconstruction and generation quality, both outperforming vanilla VQGAN, while the reconstruction quality of VQGAN (CLIP) trails that of vanilla VQGAN. One contributing factor is that different learning objectives used to train VFMs influence their ability to extract detailed and semantic features from images, thereby affecting downstream image reconstruction and generation quality. As evidence, both DINOv2 [9] and SigLIP2 [45] employed a masked prediction objective to optimize their VFMs, whereas CLIP [36] did not.

## 3.3 VFMTok

Building upon the semantically rich features provided by vision foundation models—typically structured as regular 2D grids—we introduce VFMTok, a region-adaptive tokenizer that identifies semantically coherent, irregular local regions to produce region-adaptive tokens. These tokens are sequentially quantized for decoding, with tailored learning objectives to enhance performance. In the following, we detail the architecture of VFMTok, including its region-adaptive token generation module and dedicated decoder for both image and feature reconstruction. We further describe the training objectives, which combine a pixel-level reconstruction loss for image synthesis with a feature reconstruction loss that preserves the semantic content of the foundation model's representations.

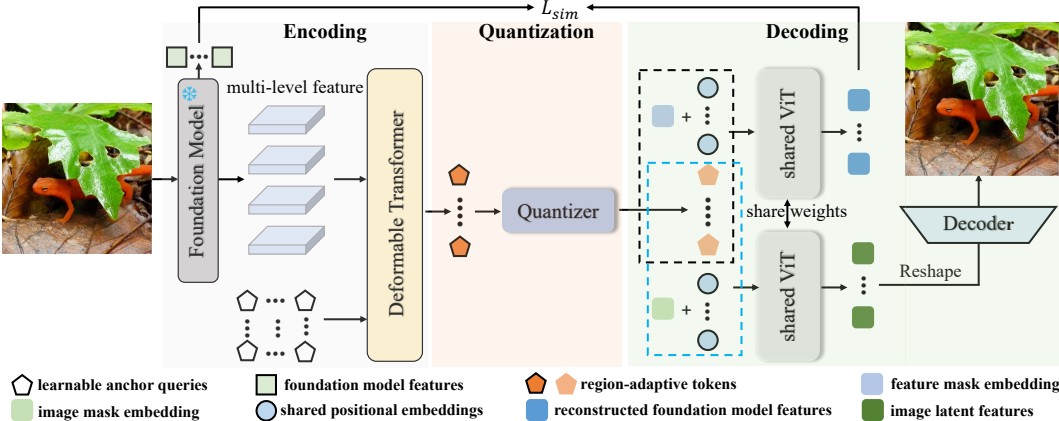

Figure 2: The framework of VFMTok. VFMTok utilizes a frozen VFM to extract multi-level image features. A deformable Transformer then processes these features with learnable grid queries to generate region-adaptive tokens. After quantization, these tokens are fed into a shared ViT for dual reconstruction: 1) VFM features, targeting similarity with the VFM's last-layer outputs, and 2) image latent features, which are reshaped to a 2D grid and decoded into pixels.

**Region-adaptive Token Generation.** Following our pilot study, we utilize a frozen pre-trained vision foundation model (VFM) to encode an input image $I$ into latent embeddings $\mathcal{F}$. Since features extracted from VFMs contain rich details in shallower layers and high-level semantics in deeper layers [7, 27, 28]—both of which are critical for image reconstruction—we extract multi-level features $\mathcal{F}_m$ from the VFM. These multi-level features are then projected to a uniform embedding dimension using a two-layer MLP. Next, as shown in Fig. 2, based on the multi-level features $\mathcal{F}_m$, we introduce a region-adaptive sampling mechanism using deformable cross-attention layers [8, 62]. A set of learnable anchor queries, initialized as a 2D grid, are iteratively refined through multiple deformable attention layers. In each layer, an anchor query predicts sampling offsets for each VFM feature level via a linear layer, enabling sampling from irregular, data-dependent positions. These sampled features are then weighted using attention scores—computed through another linear layer—and aggregated to update the query. Through this process, the anchor queries are progressively refined to capture semantically coherent, region-specific information. The final refined queries are referred to as region-adaptive tokens $Z_r$, which are subsequently quantized into discrete tokens $\tilde{Z}_r$. Compared to a fixed 2D feature grid, VFMTok adaptively aggregates features from semantically coherent, irregular regions. This substantially reduces redundancy, enabling the use of fewer tokens while maintaining superior image reconstruction and generation performance. As shown in Tab. 1, just 256 semantically rich tokens from VFMTok are sufficient to achieve high-fidelity reconstruction and generation.

**Vector Quantization.** Once the continuous region-adaptive tokens $Z_r$ are obtained, a quantizer $Q_c(\cdot)$ is applied to discretize them into region-adaptive discrete tokens $\tilde{Z}_r$. Given that the design of the codebook plays a critical role in the performance of an image tokenizer, we follow the practices in [42, 51] by applying $\ell_2$-normalization to the codebook vectors. Additionally, we adopt a low-dimensional embedding space with a large codebook size to enhance both reconstruction quality and codebook utilization following [42, 51].

**Decoder of VFMTok for Image and VFM Feature Reconstruction.** After de-quantization, the region-adaptive tokens $\tilde{Z}_r$ are used for image reconstruction. Since these tokens represent irregular,

region-level features, decoding them into a regular 2D image grid requires alignment. To achieve this, we introduce a set of mask tokens $M_\text{I}$, representing a 2D feature map of size $H_m \times W_m$ with channel dimension $C$. The mask tokens are initialized by replicating a single learnable token $H_m \times W_m$ times. Position embeddings $E$, encoding spatial locations, are then added to form position-aware masked tokens. Next, the de-quantized region-adaptive tokens $\tilde{Z}$ are concatenated with $M_\text{I}$, and the combined sequence is processed by a lightweight Transformer $\mathcal{E}_\text{ViT}(\cdot)$, which propagates information from the region-adaptive tokens to the masked image tokens. This Transformer employs *causal* self-attention, aligning its latent space with the structure of autoregressive models. Following DINOv2 [9], we further enrich the input sequence by appending a [CLS] token and several register tokens to improve representation learning and capture global context—though these are not used for reconstruction. The output of this Transformer is a refined set of mask tokens $\mathcal{F}_\text{I}$ representing a regular 2D grid structure. These are reshaped into a spatial grid and passed into a decoder $\mathcal{D}(\cdot)$ to reconstruct the image.

To preserve the semantic integrity of the VFMTok tokens, we also reconstruct high-level features (specifically, from the final layer) of the vision foundation model (VFM). This process mirrors image reconstruction: a new set of mask tokens $\mathbf{M}_f$ is initialized and augmented with positional embeddings $E$, shared with those used in image reconstruction. The concatenation of $Z_r$ and $\mathbf{M}_f$ is then processed by the same shared Transformer $\mathcal{E}_\text{ViT}(\cdot)$ to produce $\mathcal{F}_\text{P}$, the reconstructed high-level VFM feature map. By sharing $\mathcal{E}_\text{ViT}(\cdot)$ between image and feature reconstruction, we reduce the model's parameter footprint while ensuring the semantic fidelity of the latent tokens. Note that the VFM feature reconstruction is only applied during tokenizer training.

**Training Objective.** For tokenizer optimization, we follow the training objectives of VQGAN [13, 42], with one key modification: we replace its original discriminator with a pre-trained DINOv1-S [3] model. This substitution provides adversarial training guidance in a more semantically meaningful way compared to conventional discriminators such as PatchGAN [19], and we find it consistently improves reconstruction quality. In addition to image reconstruction, we incorporate a feature reconstruction objective by computing the cosine similarity loss between the reconstructed features and the corresponding frozen features from the pre-trained vision foundation model (VFM). The overall training loss is defined as: $\mathcal{L} = \alpha \cdot \mathcal{L}_\text{AE} + \lambda \cdot \mathcal{L}_\text{sim}$, where $\mathcal{L}_\text{AE}$ denotes the image reconstruction loss and $\mathcal{L}_\text{sim}$ is the feature reconstruction loss. In our experiments, we set both $\alpha$ and $\lambda$ to 1.

## 3.4 Autoregressive Image Generation

Once VFMTok is trained, the optimized discrete region-adaptive tokens $\tilde{Z}_r$ can be integrated into an autoregressive (AR) Transformer, where they are generated sequentially via a next-token prediction mechanism, conditioned on a class or text embedding $c$. The generated tokens are then passed through the Transformer encoder $\mathcal{E}_\text{ViT}(\cdot)$ to produce latent image features $\mathcal{F}_\text{I}$, which are subsequently decoded into images using the decoder $\mathcal{D}(\cdot)$. In the AR model, the region-adaptive tokens $\tilde{Z}_r$ are augmented with positional embeddings—specifically 2D Rotary Position Embeddings (RoPE) [41]—to better capture their spatial locality and structure.

## 4 Experiment

### 4.1 Setup

**Image Tokenizer.** In the main experiment, we initialize the encoder of VFMTok with a frozen pre-trained DINOv2-L [9]. Considering its composition of 24 Transformer layers, we extract features from the 6th, 12th, 18th, and 24th layers to create multi-level features. Consistent with [42, 55], we set the codebook vector dimension of the quantizer to 12 with a codebook size of 16384, to achieve a better reconstruction quality and efficient codebook utilization. Meanwhile, VFMTok utilizes 256 tokens to represent an image. Besides, the depth of the Transformer is set to 6 (following [62]). The model is trained on the ImageNet [10] training set and evaluated on its validation set.

Given that the resolution of vision foundation models (VFMs) [9, 20, 36, 45, 57] is typically $336 \times 336$, while VFMTok represents images with fixed 256 tokens by default, it's comparable to vanilla tokenizers [13, 42, 61]. Thus, we train the tokenizer on $336 \times 336$ images. Except this, we keep the training settings unchanged as LlamaGen [42]. During evaluation, the reconstructed images of $336 \times 336$ are resized to $256 \times 256$ for evaluation, which is consistent with the evaluation procedure in LlamaGen [42].

**Class-conditional Autoregressive Image Generation.** Following the generation procedure in LlamaGen [42], the AR models first generate images of $336 \times 336$ and then resize them to $256 \times 256$ for evaluation. Considering computational costs, we set the training duration based on the number of models' parameters. Models with fewer than 1B parameters are trained for 300 epochs, while the remaining models are trained for 200 epochs. Beyond the resolution and training duration, all models are trained with the same settings as LlamaGen [42]. Furthermore, we also incorporated the same VFMTok into the RAR [54] autoregressive generation framework, with all training settings remaining consistent with RAR [54]. Additionally, in our experiments, AR generation is conducted with both classifier-free guidance (CFG) and a CFG-free protocol.

**Evaluation metrics.** To evaluate image generation performance, we use Fréchet inception distance (FID) [18] and Inception Score (IS) [39] as the main metrics to measure the generation quality of different models. In addition, sFID, Precision, and Recall [23] are also reported following [42].

### 4.2 Main Results

**Image Reconstruction.** We compare VFMTok against representative 2D image tokenizers, VQGAN [13], MaskGiT [4], ViT-VQGAN [51], and 1D tokenizer, TiTok [55]. As shown in Tab. 2, our tokenizer represents an image with just 256 tokens, considerably fewer than some counterparts. For instance, the VQGAN variant LlamaGen [42] uses 576 tokens, while VQGAN [13] and ViT-VQGAN [51] even utilize up to 1024 tokens. Despite this efficiency, VFMTok achieves a strong rFID of **0.89**, and further demonstrates 100% utilization of the codebook.

The rIS score of **215.4** achieved by VFMTok significantly outperforms other methods, *e.g.*, TiTok [55] and the VQGAN series. The rIS metric quantifies the KL-divergence between the original label distribution and the logit distribution of reconstructed images after softmax normalization, thereby measuring the semantic consistency between reconstructed and original images. The higher rIS confirms VFMTok is more effective at preserving semantic consistency during reconstruction.

Table 2: Comparison with other image tokenizers. $^{oim.}$ indicates trained on OpenImages [22]. $\mathcal{Q}_c/\mathcal{Q}_P$ denotes the codebook usage in contextual and patch-level quantizers, respectively.

| Method | $f$ | Tokenizer Setup Size | Dim. | #Tok. | Image Recon. rFID↓ | rIS↑ | Usage (%)↑ $\mathcal{Q}_C$ | $\mathcal{Q}_P$ |
|---|---|---|---|---|---|---|---|---|
| TiTok [55] | – | 8192 | 64 | 256 | 1.05 | 191.5 | 100 | – |
| ImageFolder [26] | – | 32768 | 32 | 286 | **0.69** | 201.5 | 100 | – |
| VQGAN$^{oim.}$ [13] | | 256 | 4 | | 1.44 | – | – | – |
| VQGAN [13] | 8 | 8192 | 256 | 1024 | 1.49 | – | – | – |
| ViT-VQGAN [51] | | 8192 | 32 | | 1.28 | 192.3 | – | 95.0 |
| VQGAN$^{oim.}$ [13] | | 16384 | 4 | | 1.19 | – | – | – |
| VQGAN [13] | | 1024 | 256 | 256 | 7.94 | – | – | – |
| MaskGiT [4] | 16 | 1024 | 256 | 256 | 2.28 | – | – | – |
| VAR [43] | | 4096 | 32 | 680 | 0.92 | 196.0 | – | 100 |
| RQ-VAE [24] | 32 | 16384 | 256 | 1024 | 1.83 | – | – | – |
| VQGAN [13] | | | 256 | 256 | 4.98 | – | – | – |
| VQGAN [42] | 16 | 16384 | 8 | 441 | 1.21 | 189.1 | – | 99.2 |
| VQGAN [42] | | | | 576 | 0.95 | 197.3 | – | 99.7 |
| **VFMTok (Ours)** | – | 16384 | 12 | 256 | 0.89 | **215.4** | 100 | – |

**Class-conditional Image Generation.** We evaluate VFMTok on vanilla autoregressive models – LlamaGen [42], and advanced generative model – RAR [54] with different scales by conducting $256 \times 256$ class-conditional image generation task on ImageNet [10], where comparing them with the mainstream generation models, including diffusion models (Diff.) [30, 34, 50, 60], BERT-style masked-prediction models (Mask.) [4], and AR generation models (AR) [13, 24, 38, 42, 51, 55].

As shown in Tab. 3, our models exhibit competitive performance across all metrics compared to mainstream image generation models. Notably, VFMTok beats BERT-style models [4] in terms of gFID without the requirement of complicated sampling tuning. With comparable or even fewer parameters, our method surpasses most AR generative models [13, 24, 38, 51, 55] in both gFID and gIS metrics. Under the same training setting, VFMTok surpasses LlamaGen [42] by significant gFID gains and notable gIS improvements. Specifically, VFMTok-B outperforms LlamaGen-B [42] with gains of **2.56** in gFID and **69.7** in gIS. Besides, our VFMTok-L model achieves a gFID of **2.75** at 300 epochs, also obtaining a gain of **22.7** in gIS. Notably, when compared with LlamaGen-3B with 3B parameters, our VFMTok-XXL achieves even better generation performance with less than half the number of parameters and fewer training iterations. Futhermore, when VFMTok is incorporated into RAR [54], it achieves a generative performance with gFID of **1.36**, which is the state-of-the-art generation performance at present. Additionally, class-conditional image generation results are visualized in the Appendix.

Table 3: Class-conditional image generation quality estimated on ImageNet [10] validation benchmark. † indicates it is implemented by us, and '-re' indicates using rejection sampling.

| Type | Method | #Epoch | #Para. | #Tok. | Generation w/ CFG | | | | | Generation w/o CFG | | | | |
|---|---|---|---|---|---|---|---|---|---|---|---|---|---|---|
| | | | | | gFID | sFID | gIS | Pre. | Rec. | gFID | sFID | gIS | Pre. | Rec. |
| Diff. | MaskDiT [60] | 1600 | 675M | | 2.28 | 5.67 | 276.6 | 0.80 | 0.61 | 5.69 | 10.34 | 177.9 | 0.74 | 0.60 |
| | DiT [34] | 1600 | 675M | _ | 2.27 | 4.60 | 278.2 | 0.83 | 0.57 | 9.62 | 6.85 | 121.5 | 0.67 | 0.67 |
| | SiT [30] | 1600 | 675M | | 2.06 | 4.50 | 270.3 | 0.82 | 0.59 | 8.61 | 6.32 | 131.7 | 0.68 | 0.67 |
| | FasterDIT [50] | 400 | 675M | | 2.03 | 4.63 | 264.0 | 0.81 | 0.60 | 7.91 | 5.45 | 131.3 | 0.67 | 0.69 |
| Mask. | MaskGiT [4] | 555 | 227M | 256 | – | – | – | – | – | 6.18 | – | 182.1 | 0.80 | 0.51 |
| | MaskGiT-re | | | | 4.02 | – | 355.6 | – | – | – | – | – | – | – |
| AR | VAR [43] | 350 | 310M | 680 | 3.30 | – | 274.4 | 0.84 | 0.51 | – | – | – | – | – |
| | TiTok-B† [55] | 300 | 111M | 256 | 6.76 | 7.82 | 175.3 | 0.85 | 0.43 | 19.6 | 57.9 | 7.54 | 0.64 | 0.60 |
| | TiTok-L† [55] | | 343M | | 4.03 | 6.93 | 219.5 | 0.84 | 0.52 | 11.4 | 88.8 | 7.14 | 0.68 | 0.64 |
| | LlamaGen-B | 300 | 111M | 576 | 6.09 | 7.24 | 182.5 | 0.85 | 0.42 | 32.2 | 11.84 | 39.9 | 0.57 | 0.61 |
| | LlamaGen-L | | 343M | | 3.07 | 6.09 | 256.1 | 0.83 | 0.52 | 19.1 | 8.67 | 64.3 | 0.61 | 0.67 |
| | LlamaGen-XXL | | 1.4B | | 2.34 | 6.00 | 253.9 | 0.81 | 0.60 | 14.6 | 8.69 | 86.3 | 0.63 | 0.68 |
| | LlamaGen-3B | | 3.1B | | 2.19 | 5.97 | 263.3 | 0.82 | 0.58 | 9.38 | 8.24 | 112.9 | 0.69 | 0.67 |
| | RAR-L [54] | | 461M | 256 | 1.70 | – | 299.5 | 0.82 | 0.58 | 6.72 | 5.56 | 129.2 | 0.74 | 0.61 |
| | RAR-XL [54] | 400 | 955M | | 1.50 | – | 306.9 | 0.80 | 0.62 | 4.62 | 5.27 | 158.3 | 0.77 | 0.62 |
| | RAR-XXL [54] | | 1.5B | | 1.48 | – | 326.0 | 0.80 | 0.63 | 3.85 | 5.18 | 176.2 | 0.78 | 0.61 |
| | VFMTok-B | 300 | 111M | 256 | 3.43 | 5.88 | 252.2 | **0.85** | 0.53 | 3.09 | 5.67 | 173.6 | 0.80 | 0.58 |
| | VFMTok-L | | 343M | | 2.75 | 5.58 | 278.8 | 0.84 | 0.57 | 2.11 | 5.46 | 230.1 | 0.82 | 0.60 |
| | VFMTok-XXL | 200 | 1.4B | | 2.19 | 5.53 | 278.0 | 0.83 | 0.60 | 1.95 | 5.65 | 259.3 | 0.82 | 0.62 |
| | VFMTok-3B | 200 | 3.1B | | 2.07 | 6.23 | 280.4 | 0.81 | **0.62** | 2.04 | 5.43 | 267.6 | 0.82 | 0.61 |
| | RAR-L(VFMTok) | | 461M | | 1.44 | 6.03 | 312.8 | 0.78 | 0.66 | 2.02 | 5.51 | 210.4 | 0.79 | 0.63 |
| | RAR-XL(VFMTok) | 400 | 955M | 256 | 1.38 | 5.86 | 310.2 | 0.78 | 0.65 | 1.74 | **5.33** | 233.0 | 0.80 | 0.63 |
| | RAR-XL(VFMTok) | | 1.5B | | **1.36** | 5.85 | 301.3 | 0.78 | 0.66 | **1.65** | 5.55 | 253.7 | 0.80 | 0.63 |
| | VFMTok-L(SigLIP2) | 300 | 343M | | 2.69 | **5.31** | 273.4 | 0.84 | 0.56 | 2.11 | 5.39 | 225.6 | 0.81 | 0.60 |
| | VFMTok-XXL(SigLIP2) | 200 | 1.4B | 256 | 2.16 | 5.45 | 272.0 | 0.83 | 0.60 | 1.98 | 5.53 | 265.3 | 0.82 | 0.62 |
| | VFMTok-2B(SigLIP2) | 200 | 2.2B | | 2.17 | 5.43 | 281.4 | 0.83 | 0.60 | 1.98 | 5.41 | **269.7** | 0.82 | 0.62 |

Furthermore, we conducted experiments by **removing classifier-free guidance (CFG)**. Remarkably, the generation results without CFG show that most evaluation metrics—such as sFID, Precision, and Recall—remain comparable to those obtained with CFG. While gIS experiences a slight decline, gFID improves compared to its CFG-enabled counterpart. Similar trends are observed when VFM-Tok's encoder is replaced with other frozen pre-trained vision foundation models like SigLIP2 [45]. These results demonstrate that our method supports high-fidelity autoregressive image generation even without CFG, which significantly accelerates inference. In contrast, baseline methods suffer substantial performance degradation without CFG—for example, LlamaGen-3B model sees gFID worsen to **9.38**, whereas our 1.4B model VFMTok-XXL achieves a gFID of **1.95** without CFG.

### 4.3 Ablation Study and More Analysis

**Component study.** To assess the contribution of each proposed component to image reconstruction and synthesis, we conduct a step-by-step component analysis using a baseline tokenizer built on vanilla VQGAN [42]. We incrementally add the following components: (1) replace the VQGAN encoder with a frozen pre-trained foundation model (DINOv2-L [9]); (2) introduce learnable queries and a deformable attention for region-adaptive tokenization, using only single-level features from the final layer; (3) incorporate multi-level features to enrich representations with both low-level detail and high-level semantics; and (4) add a feature reconstruction objective based on pre-trained VFM outputs. After training each tokenizer, we integrate it with our AR generation model, VFMTok-L, for autoregressive image synthesis. Both the tokenizer and AR model are trained for 50 epochs. Additionally, we perform linear probing on the [CLS] token, following the MAE [15] protocol.

As shown in Tab. 4, replacing VQGAN's encoder with a frozen pre-trained vision foundation model yields reconstruction and generation performance on par with a VQGAN trained specifically for

visual reconstruction using 576 tokens. This substitution also significantly enhances the semantic quality of the tokenizer's representations. To further improve token efficiency, we introduce region-adaptive tokenization using deformable attention to exploit the spatial redundancy inherent in regular 2D grid features. This reduces the number of visual tokens to 256. However, this performance gain comes at a cost: reconstruction and generation quality degrade slightly due to two factors: (1) fewer visual tokens limit representational capacity, and (2) the absence of explicit supervision hinders the effective optimization of the region-adaptive tokens. To address this, we incorporate multi-level feature extraction, which improves the reconstruction capability by leveraging both low- and high-level information. However, without additional guidance, the semantic consistency of the learned tokens may still degrade. Finally, we introduce a pre-trained feature reconstruction objective, which significantly boosts both image reconstruction and generation quality. This objective encourages alignment with semantic features from the frozen VFM and effectively balances the contributions of low- and high-level features to the contextual tokens—thereby preserving semantic fidelity.

With these three key components—(1) deformable attention for region-adaptive tokenization to reduce redundancy, (2) multi-level features for enhanced reconstruction, and (3) feature reconstruction loss for semantic alignment—VFMTok produces compact, semantically rich, and efficient tokens. Using only 256 tokens, VFMTok outperforms its VQGAN baseline with 576 tokens in reconstruction quality, generative performance, and semantic representation. Supplemental ablations are discussed in the Appendix.

Table 4: Ablation study on VFMTok's components.

| Setup | Image Recon. | | | Usage | AR Gen. | | L.P. |
|---|---|---|---|---|---|---|---|
| | #Tok. | rFID↓ | rIS↑ | $\mathcal{Q}_C$ ↑ | gFID↓ | gIS↑ | (%) |
| VQGAN | 576 | 0.95 | 197.3 | 99.7% | 3.71 | 228.3 | 23.1 |
| + Frozen VFM | | 0.99 | 206.3 | 100% | 3.69 | 267.5 | 56.4 |
| + Region Adapt. | | 1.20 | 199.0 | | 3.98 | 241.6 | 41.5 |
| + Multi-level Feat. | 256 | 0.92 | 199.5 | 100% | 3.71 | 251.1 | 22.7 |
| + Reconstruct Feat. | | **0.89** | **215.4** | | **3.42** | **277.3** | **69.4** |
| - Frozen VFM | 256 | 0.95 | 196.3 | 100% | 3.73 | 248.7 | 59.1 |

**Convergence and efficiency analysis.** Beyond above analysis, we experiment VFMTok with a randomly initialized encoder instead of a pre-trained VFM with other components remaining unchanged. As shown in Tab. 4 (last row), its reconstruction quality dropped to the level of VQGAN. Meanwhile, both its semantic representation capability and generation performance also decreased. This indicates a frozen VFM benefits tokenizer training as it provides a latent space advantageous for image reconstruction and generation. Besides, those semantic-rich, structured latent tokens accelerate AR model training convergence. As evidenced in Fig. 1(b), VFMTok enables AR models to achieve a **3×** speedup in convergence compared to VQGAN. Moreover, an AR model's generation time is quadratically proportional to the number of tokens. At the same resolution, VFMTok uses approximately half the tokens for image representation compared to counterparts like DINOv2-VQGAN and CLIP-VQGAN. Consequently, VFMTok achieves a **4×** generation speedup over these counterparts depicted in Tab. 1. This acceleration can be further enhanced with CFG-free generation.

## 5 Conclusion

In this work, we have demonstrated that frozen pre-trained vision foundation models (VFMs)—ranging from self-supervised to language-supervised– are sufficient for high-quality image reconstruction and generation. To fully exploit their potential while addressing the redundancy inherent in 2D feature grids, we introduce VFMTok, a novel image tokenizer that incorporates region-adaptive tokenization to enhance token efficiency. By reducing feature redundancy, integrating multi-level feature representations, and introducing a semantic-preserving feature reconstruction objective, VFMTok yields a compact and semantically rich latent space. This facilitates high-quality image reconstruction and generation, accelerates convergence in autoregressive (AR) models, and enables efficient, high-fidelity, classifier-free (CFG-free) image synthesis—without the need for additional heuristics. Furthermore, the reduced number of tokens significantly lowers the computational cost of AR inference, making the approach both scalable and effective. Looking forward, the rich semantic structure of the learned latent space offers exciting potential for extending this work toward unified visual generation and understanding.

# 6 Acknowledgments

This work has been supported by the National Key R&D Program of China (Grant No. 2022YFB3608300), Hong Kong Research Grant Council - Early Career Scheme (Grant No. 27209621), General Research Fund Scheme (Grant No. 17202422, 17212923, 17215025) Theme-based Research (Grant No. T45-701/22-R) and Shenzhen Science and Technology Innovation Commission (SGDX20220530111405040). Part of the described research work is conducted in the JC STEM Lab of Robotics for Soft Materials funded by The Hong Kong Jockey Club Charities Trust. We are deeply grateful to Lufan Ma for the contribution in polishing up this paper. We would also appreciate Tong Yang for providing the DINO discriminator script.

# 7 Author Contribution Statement

X.Q. proposed the initial concept of region-adaptive quantization. Based on this, A.Z. built the VFMTok, conducted the overall experiments, and led the writing of the initial draft. X.W., X.Q., and C.M. were deeply involved in the project progress and manuscript writing. X(iangyu).Z., X(uanyang).Z., G.Y., and T.W., provided sufficient computational resources. X(uanyang).Z. joint discussion where suggested ablation studies along with T.W., and discussed the writing of the draft. All authors contributed critical feedback, shaping the research, analysis, and the final manuscript.

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
