# OpenReview forum: "Vision Foundation Models as Effective Visual Tokenizers for Autoregressive Generation"
_NeurIPS.cc/2025/Conference — NeurIPS 2025 poster_

### Official Review · Reviewer_vMYn · 2025-06-29

**Clarity:** 3
**Significance:** 2
**Originality:** 2
**Rating:** 4
**Confidence:** 5

**Summary:**

This paper proposes using vision foundation models (VFMs) such as CLIP and DINOv2 as part of an image tokenizer to be used for training visual autoregressive generative models. In particular, they leverage frozen VFMs as the encoding part of a VQ-VAE style discrete tokenization framework. Given an image, VFM extracts features from it at several levels (or layers). Afterwards, these features are resampled by a Transformer with deformable cross-attention layers to reduce redundancy and then quantized. The quantized features are then fed as input to another lightweight Transformer whose goal is to both predict the original VFM features (to preserve the semantics) as well as image latent features. The latter is fed to the decoder of VQ-VAE to reconstruct the input image. After the tokenizer is trained, the resulting discrete tokens can be used to train an autoregressive Transformer for class-conditioned image generation, which they did on ImageNet to show the effectiveness of the method.

**Questions:**

Please see the strengs & weaknesses for details.

- How do you compare against recent work leveraging VFMs as part of a visual generation pipeline? Both conceptually and empirically.

- What is the performance of different VFMs within the proposed framework? What are their characteristics leading to varying performance?

- Could you clarify the questions regarding evaluation setup? First of all, I think you should also report total training FLOPs, as right now it's a bit unclear as you are operating at a different resolution. Second, some reported numbers are confusing. For example, TiTok reimplementation results are much worse than the original numbers reported, could you explain why?

- What are the contributions of different aspects of the method? For example, I would be curious to see 1) whether other VFMs show similar trends and 2) what if you simply extract multi-level features from frozen VFMs without region-adaptive tokenization and with & without reconstructing the features. I also found it interesting that linear proping accuracy of the randomly initialized encoder is much better than vanilla VQGAN (15.0 vs 59.1). Could it be because of multi-level features and the reconstructing the features as an objective? More clarifications and analysis would be nice. Could you also clarify if running the ablations with 50 epoch training leading to stable results?

- What would be the performance on other generative benchmarks such as COCO and GenEval?

- What are the limitations of the method? What are possible future directions to investigate?

**Ethical Concerns:**

["NO or VERY MINOR ethics concerns only"]

**Final Justification:**

Thanks for the further clarifications. Overall, rebuttal sufficiently addressed my concerns, thus I am happy to increase my rating to borderline accept.

I agree with fellow reviewers SUoJ and APhE that the novelty of the submission is limited which is a factor for me to not increase the rating further. Still, some of the introduced components like region adaptive quantization and the analysis made in the paper might be relevant to the community.

Rebuttal wasn't enough at addressing my question on the VFMs trained with masked prediction objective doing better than CLIP. I guess we can pose this question more generally and say: What makes a VFM a good visual tokenizer? Having a solid answer to this would have improved the significance (and rating) of this paper a lot more.

**Limitations:**

No limitation is provided. I provided some pointers on the strengths and weaknesses section on several aspects that can be improved.

**Paper Formatting Concerns:**

Some qualitative results would be nice.

**Quality:**

2

**Strengths And Weaknesses:**

Improving efficiency and effectiveness of autoregressive visual generation models is a timely topic. The paper observes a critical point about the convergence issues of VQGAN-style tokenizers and proposes using semantically rich features from VFMs. This is not a new idea, as several papers previously investigated it [A] [B] [C] [D], yet the paper further boosts performance with a region-adaptive tokenization strategy, reducing the total number of tokens required. While I view this to be an interesting direction, there are several issues that must be addressed first:

- Missing related work. As I mentioned, using VFMs as part of tokenization pipeline is not a new idea. I'd expect to see a dedicated related work section on this and contrasting the paper to those approaches.

- Importantly, recent approaches like REPA showed promising results in terms of improving convergence speed by aligning representations to a VFM, namely DINOv2. FlexTok used this idea to improve the quality and efficiency of discrete image tokenizers. Since this paper claims similar gains, a direct analysis and comparison would be useful to those REPA-based approaches.

- I found the evidence provided in L181 weak. Why do we think VFMs trained with masked prediction objective do better than CLIP? What about MAE then? I think, overall, a more clear and focused analysis on the performance of different VFMs is needed to make the papers claims stronger.

- Evaluation setting and some evaluation results require further clarifications. First of all, I think you should also report total training FLOPs, as right now it's a bit unclear as you are operating at a different resolution. Second, some reported numbers are confusing. For example, TiTok reimplementation results are much worse than the original numbers reported, could you explain why?

- Ablations are interesting, I would be curious to see 1) whether other VFMs show similar trends and 2) what if you simply extract multi-level features from frozen VFMs without region-adaptive tokenization and with & without reconstructing the features. I also found it interesting that linear probing accuracy of the randomly initialized encoder is much better than vanilla VQGAN (15.0 vs 59.1). Could it be because of multi-level features and the reconstructing the features as an objective? More clarifications and analysis would be nice. Could you also clarify if running the ablations with 50 epoch training leading to stable results?

- While I acknowledge ImageNet is a typical benchmark to show generative capabilities, I would be curious to see the performance on other benchmarks such as COCO and GenEval. In particular, as the paper claims, I would be curious if the semantic features from those VFMs are helpful in more challenging settings.

- A dedicated limitation section is also needed.

[A] Representation Alignment for Generation: Training Diffusion Transformers Is Easier Than You Think

[B] Making LLaMA SEE and Draw with SEED Tokenizer

[C] Emu: Generative Pretraining in Multimodality

[D] FlexTok: Resampling Images into 1D Token Sequences of Flexible Length

---

> ### Author Rebuttal · Authors · 2025-07-31
>
> Dear reviewer `vMYn`,
>
> Thank you for your thorough review of our paper and detailed suggestions for improvement. Your questions and concerns are answered below:
>
> > **W1:** Using VFMs as part of tokenization pipeline is not a new idea. A dedicated related work section is needed.
>
> Thanks for your suggestion. We note that our motivation is to show that **frozen** VFM features **alone** are sufficient to build an image tokenizer, while all previous works use VFM for other purposes or in other ways. To make a clear comparison, we provide a table on the purpose of using VFMs below:
>
> |Approach|Encoder initialization| Feature regularization|Conditional feature generation|Codebook initialization|
> |:---|:---:|:---:|:---:|:---:|
> |REPA[A]| | $\checkmark$| | |
> |SEED[B]| |	| $\checkmark$| |
> |EMU[C] | | | $\checkmark$|	|
> |FlexTok[D]| |$\checkmark$ || |
> |Lightning-DiT| | $\checkmark$| | |
> |SoftVQ-VAE| | $\checkmark$| | |
> |VILA-U| $\checkmark$| | | |
> |SemHiTok| $\checkmark$|$\checkmark$|||
> |TokenFlow| $\checkmark$|$\checkmark$|||
> |VQGAN-LC| | | | $\checkmark$|
>
>  Table (a). The primary purpose of VFM.
>
> |Tokenizer|\#code.|\#tok.|rFID$\downarrow$|rIS$\uparrow$|gFID$\downarrow$|gIS$\uparrow$|
> |:---|:---:|:---:|:---:|:---:|:---:|:---:|
> |VQGAN-LC[58]| 100000|256|2.62|--|15.4|--|
> |VFMTok-L|16384| 256|**0.89**|**215.4**|**2.11**|**230.1**|
>
> Table  (b). Performance comparsion between VQGAN-LC[58] and VFMTok.
>
> - As shown in the Table.(a), concerning the purpose of VFMs, prior methods have primarily used VFM for: 1) encoder initialization (e.g., EMU[C], VILA-U, SemHiTok, TokenFlow); 2) feature regularization (e.g., REPA[A], Lighting-DiT, SoftVQ-VAE, Flextok[D]); 3) conditional feature generation for diffusion models (e.g., EMU, SEED[B]); and 4) codebook initialization (e.g., VQGAN-LC).
> - Moreover, approaches that utilize a VFM to initialize the tokenizer's encoder (such as EMU, VILA-U, and TokenFlow) all explicitly update the VFM's parameters during training.
> - Empirically, the most related work is VQGAN-LC[58], which investigates initializing its codebook by heuristically clustering CLIP patch features. Similarly, its parameters in VFM also remain frozen during the training phase. Nevertheless, its performance in image reconstruction and generation is substantially lower than that of VFMTok. As indicated in the Table.(b) above.
> - In conclusion, all of them cannot build a tokenizer with frozen VFMs solely. We will incorporate the discussion into the manuscript for a clearer relation with related works.
>
> > **W2:** REPA showed improved convergence by aligning to DINOv2. FlexTok used REPA to improve discrete image tokenizers. A comparison is needed.
>
> **Comparison with REPA.** Here we convert VFMTok into a continuous VAE (VFMVAE) and integrate it into SiT/XL. Image reconstruction and generation performance are presented below. For simplicity and fairness, images are generated without classifier-free guidance(CFG).
>
> || rFID($\downarrow$)| rIS($\uparrow$)| gFID($\downarrow$)| gIS($\uparrow$)|
> |:---| :---: | :---: | :---: | :---: |
> |SiT-XL|0.61|212.1|8.30|131.7|
> |SiT-XL (REPA)|0.61|212.1|5.90|157.8|
> |SiT-XL (VFMVAE)|**0.46**|**224.4**|**3.08**|**175.0**|
>
> **Comparison with FlexTok.** VFMTok achieves better reconstruction and generation quality with approximately half of the parameters in the AR generation model.
>
> ||\#code.|\#tok.|rFID($\downarrow$)| rIS($\uparrow$)|AR params.|gFID($\downarrow$)| gIS($\uparrow$)|
> |:---|:---|:---|:---|:---|:---|:---|:---|
> |FlexTok|64000|256|1.08|--|1.3B|2.45|258.3|
> |VFMTok (XXL)|16384|256|**0.89**|**215.4**|**775M**|**2.41**|**276.8**|
>
> > **W3:** Why do we think VFMs trained with masked prediction objective do better than CLIP? What about MAE then?
>
> - In this setup, the encoder of VFMTok is substituted with MAE-L. Training setup: VFMTok (50 epochs), AR model, VFMTok-L (100 epochs). As shown in the table below, while the generation and reconstruction quality of VFMTok (MAE) does not match that of VFMTok (CLIP), it demonstrates a significant improvement over the baseline. Moreover, its reconstruction performance is better than that of VQGAN (CLIP).
> - This suggests that VFMTok, integrating pre-trained models with a masked prediction objective, is advantageous for image reconstruction. Nevertheless, due to factors such as the VFM's model size, the volume of training data, and time constraints, VFMTok (MAE) is not yet comparable to VFMTok (CLIP).
> - We will investigate this further and provide a more complete discussion in the paper.
>
> ||\#code.|\#tok.|rFID($\downarrow$)| rIS($\uparrow$)|gFID($\downarrow$)| gIS($\uparrow$)|
> |:---|:---:|:---:|:---:|:---:|:---:|:---:|
> |VQGAN[43]|16384|576|0.95|197.3|3.71|228.3|
> |VQGAN (CLIP)|16384|576|1.47|182.0|3.45|221.2|
> |VFMTok (CLIP)|16384|256|0.99|**200.1**|3.40|**274.7**|
> |VFMTok (MAE)|16384|256|**0.98**|191.8|**3.34**|258.9|
>
> > **W4.1:** Total training FLOPs should be reported, as currently you operate at a different resolution.
>
> - We presented the required training computation and throughput for VFMTok below.
> - Thanks to the use of frozen VFM features, these can be cached, and the trainable module in the encoder is only a lightweight deformable sampler. Together, the training of our tokenizer is also substantially faster.
> - Besides, we have also included 256x256 generation experiments in Appendix Table A1 for reference.
>
> | Tokenizer     | Resolution | Encoding GMac | Decoding GMac | Training throughput ($\uparrow$,img/sec) |
> | :-- | :---:|    :---:      |     :---:     |      :---:        |
> |VQGAN[43]|  256 |    69.36      |     126.63    |      205.46       |
> | VFMTok  |  256 |    9.39       |     104.24    |      286.43       |
> | VQGAN[43]|  384 |    156.62     |     285.67    |      89.84       |
> | VFMTok|  336 |    9.39       |     176.08    |      183.67       |
>
> > **W4.2:** TiTok reimplementation results require further clarification.
>
> We used the official TiTok weights, but integrated with an AR model (LlamaGen) for fair comparison with other methods instead of the original MaskGiT.  We’ll revise the text for clarity.
>
> > **W5.1:** Whether other VFMs show similar trends?
>
> Other VFM, such as VFMTok (SigLIP), also show a similar trend, which is presented in Table A6 in the Appendix.
>
> > **W5.2:** What about w/o region-adaptive tokenization, and w/ & w/o reconstructing the features?
>
> Below, we show the image reconstruction performance under the suggested settings, where the first two rows do not use region-adaptive tokenization. We train VFMTok for 50 epochs, while the AR models are trained for 100 epochs.
>
> |Tokenizer|\#code.|\#tok.|rFID($\downarrow$)| rIS($\uparrow$)|
> |:---|:---:|:---:|:---:|:---:|
> |w/o VFM recon.|16384|576|0.63|207.3|
> |w/ VFM recon.|16384|576|0.71|212.8|
> |VFMTok|16384|576|0.89|215.4|
>
> > **W5.3:** If running the ablations with 50 epoch training lead to stable results?
>
> We list the AR generation quality at both 50 and 100 epochs in the table below. The trend is consistent.
>
> |AR model|\#epochs|gFID($\downarrow$)| gIS($\uparrow$)|
> |:---|:---:|:---:|:---:|
> |w/o VFM recon|50|3.56|252.0|
> |w/ VFM recon|50|3.58|274.8|
> |VFMTok|50|**3.42**|**277.3**|
> |w/o VFM recon.|100|3.43|273.5|
> |w/ VFM recon.|100|3.20|274.0|
> |VFMTok|100|**3.09**|**274.2**|
>
> > **W5.4:** Why linear probing accuracy of the randomly initialized encoder is much better than vanilla VQGAN?
>
> - To analyze the cause of VQGAN [43]'s low linear probing accuracy, we begin by substituting the PatchGAN in VQGAN with DinoGAN. Subsequently, we incorporate the objective of feature reconstruction.
> - The table below shows that adding DinoGAN, with its semantic discriminability, remarkably improves VQGAN's accuracy. VFM feature reconstruction further boosts this accuracy.
> - We can therefore conclude that a semantically-informed loss function enhances the linear probing accuracy of VQGAN.
> |Tokenizer|top-1($\uparrow$,\%)|
> |:---|:---:|
> |VQGAN[43]|15.0|
> |VQGAN[43]+DinoGAN|40.5|
> |+Semantic recon.|58.3|
>
> > **W6:** What would be the performance on other generative benchmarks such as COCO and GenEval?
>
> Time constraints prevented us from completing the text-to-image generation task for evaluation on COCO and GenEval. We will continue to work on this and integrate VFMTok with a text-to-image AR generation model to conduct these evaluations in the future. Thanks for your suggestion!
>
> > **W7:** What are the limitations of the method? What are possible future directions to investigate?
>
> The limitations section is presented in the Appendix (page 4), and we will elaborate more on this part in the next revision.

---

> > ### Comment · Reviewer_vMYn · 2025-08-05
> >
> > Thanks for the detailed response. Regarding W5.2, what should be our takeaway? Seems like you get better rFID without VFM reconstruction and region-adaptive tokenization? Then why do we need those components? Also, what would be the corresponding generative metrics? More clarifications would be helpful to assess these results.

---

> ### Author Response · Authors · 2025-08-05
>
> 1. The table for **W5.2** need be considered along with the table in **W5.3**. From these, we can summarize the following takeaways::
>
> - More tokens in a tokenizer to represent an image can achieve a better reconstruction performance.
> - Introducing stronger semantics improves generation quality.
> - VFMTok, equipped with region-adaptive quantization, can better represent an image with fewer tokens, achieving faster convergence speed with a lesser computational budget, but obtaining better generation quality.
>
> 2. Those takeaways can be specified below:
>
> - With the region-adaptive quantization removed from VFMTok, it degraded into a vanilla VQGAN, which utilizes **576** tokens to represent an image. As the token count representing an image increases, the reconstruction quality improves accordingly. However, the rFID of the tokenizer with VFM reconstruction is slightly worse than that without it. We hypothesize that VFM reconstruction loss could adversely affect pixel-level reconstruction during training in this setup.
>
> - As shown in the table for **W5.3**, and as depicted by the gFID (an important metric that estimates image generation quality where a lower value indicated better performance), introducing stronger semantics significantly accelerates the convergence of the AR generative models and improves their generation quality.
>
> - In contrast, VFMTok, which adopts only **256** tokens (Note: there is a mistaken writing in the first table for **W5.2**. its token count should align with that in Table 2 of the manuscript.) but is equipped with the region-adaptive quantization, significantly accelerates the training convergence speed and inference speed. It also reduces computational cost and improves generation quality compared to the other 2 tokenizers containing **576** tokens. This consolidates that the region-adaptive quantization is crucial to VFMTok's efficacy.
> |AR model|\#tok.$\downarrow$|rFID$\downarrow$|rIS$\uparrow$|\#epochs|gFID$\downarrow$|gIS$\uparrow$|
> |:---|:---:|:---:|:---:|:---:|:---:|:---:|
> |w/o VFM recon|576|0.63|207.3|100|3.43|273.5|
> |w/ VFM recon|576|0.71|212.8|100|3.20|274.0|
> |VFM Tok|**256**|0.89|**215.4**|100|**3.09**|**274.2**|

---

> > ### Comment · Reviewer_vMYn · 2025-08-06
> >
> > Thanks for the further clarifications. Overall, rebuttal sufficiently addressed my concerns, thus I am happy to increase my rating to borderline accept.
> >
> > I agree with fellow reviewers SUoJ and APhE that the novelty of the submission is limited which is a factor for me to not increase the rating further. Still, some of the introduced components like region adaptive quantization and the analysis made in the paper might be relevant to the community.
> >
> > Rebuttal wasn't enough at addressing my question on the VFMs trained with masked prediction objective doing better than CLIP. I guess we can pose this question more generally and say: What makes a VFM a good visual tokenizer? Having a solid answer to this would have improved the significance (and rating) of this paper a lot more.

---

> ### Author Response · Authors · 2025-08-06
> **What makes a VFM a good visual tokenizer?**
>
> Thank you for your acknowledgement of our work, and we would like to provide more insights on the question "**What makes a VFM a good visual tokenizer?**" (for image reconstruction, generation, and possibly understanding).
>
> The main takeaways are:
> 1. MIM (masked image modeling) objectives primarily help reconstruction and generation.
> 2. MIM in pixel space helps reconstruction more, but is less beneficial for generation than MIM in latent space.
> 3. CL (contrastive learning) objectives are less helpful for reconstruction & generation, but are important for understanding abilities (eg, top-1 accuracy on ImageNet).
> 4. Best VFMs for visual tokenization are trained with both MIM in latent space and CL objectives, namely DINOv2 and SigLIP2.
>
> ****
>
> We begin with a brief discussion on the training objectives of different VFMs, and then how these VFMs perform as image tokenizers.
>
> **Training objectives of VFMs.** CLIP and SigLIP apply CL to image-text pairs, whereas DINOv1 applies a CL-like clustering objective to images. MAE performs MIM on image patches in pixel space, and iBOT predicts DINO-like cluster assignments of image patches in latent space. DINOv2 is basically DINOv1+iBOT, and SigLIP2 is also essentially SigLIP+iBOT (termed TIPS loss in their paper), omitting other regularizing losses. We hereby provide a holistic comparison of the training objectives of different VFMs below:
>
> |Method|CL|Latent MIM|Pixel MIM|
> |:---|:---:|:---:|:---:|
> |CLIP|✅|❌|❌|
> |SigLIP|✅|❌|❌|
> |**SigLIP2**|✅|✅|❌|
> |DINOv1|✅|❌|❌|
> |iBOT|❌|✅|❌|
> |**DINOv2**|✅|✅|❌|
> |MAE|❌|❌|✅|
>
> Notably, **iBOT/DINOv2/SigLIP2 are performing VQ-VAE-like codebook learning on image patches**, despite codes being soft rather than one-hot, and codebook vectors being high-dimensional (eg, 256). Following DINOv1, they learn a set of 8192 or 65536 prototypes (l2-normalized 256-dim vectors), and require two augmented versions of the same patch (or image) to have identical prototype assignments (soft codes). They also predict the soft codes of masked patches, thus performing MIM in latent space. This formulation is termed self-distillation in DINO, and then extended to patches with MIM by iBOT, and then adopted by DINOv2 and SigLIP2.
>
> The connection between Latent MIM and VQ-VAE makes it natural to convert these VFMs to tokenizers and expect good reconstruction/generation performance. Additionally, the global (image-level) CL objectives of DINOv2 and SigLIP2 also allow better high-level semantics (better performance on understanding tasks).
>
> ****
>
> To better support the discussion above, we provide an ablation on VFMs with all results at hand below. We see that:
> 1. Pixel MIM (MAE) aligns best to reconstruction objectives, thus, MAE achieves the best rFID. However, this does not provide a feature space as friendly as codebook learning (latent MIM) for generation tasks. We also do not expect strong semantics (top-1 accuracy) in this model.
> 2. CL-only models (CLIP & SigLIP) achieve best semantics (top-1 accuracy), but modest gFID and worse rFID. Without MIM objectives, they lack good modeling of low-level structures.
> 3. Latent MIM & CL co-trained methods (SigLIP2 and DINOv2) achieve the best overall performance.
>
> |Tokenizer|CL|Latent MIM|Pixel MIM|\#tok.|rFID$\downarrow$|rIS$\uparrow$|\#epochs|gFID$\downarrow$|gIS$\uparrow$|top-1(\%)$\uparrow$|
> |:---|:---:|:---:|:---:|:---:|:---:|:---:|:---:|:---:|:---:|:---:|
> |VQGAN(CLIP)|✅|❌|❌|576|1.47|182.0|100|3.45|221.2|59.5|
> |VQGAN(SigLIP)|✅|❌|❌|576|1.26|190.8|100|3.50|246.1|60.3|
> |VQGAN(SigLIP2)|✅|✅|❌|576|0.96|198.4|100|3.39|267.8|55.5|
> |VQGAN(DINOv2)|✅|✅|❌|576|0.99|206.3|100|3.34|268.6|56.4|
> |VQGAN(MAE)|❌|❌|✅|576|0.67|207.6|100|3.40|265.5|39.0|
>
> ****
> This discussion above provides a more systematic framework for what makes a VFM a good tokenizer, from the perceptive of reconstruction, generation, and understanding. We also connect their performance with specific pre-training objectives, and discover a strong association between some VFMs and VAE. Together, **our work could provide a systematic answer to what and why VFMs can be transformed to strong tokenizers, and how to achieve this.** We believe this discussion would be greatly beneficial for both pre-training and generation research.
>
> We appreciate the reviewer for raising the question and look forward to further feedback. We will also incorporate the discussion into our manuscript to strengthen the theoretical framework.
>
> ****
> We highlight the novelty of our approach as follows: VFMTok is the first to establish that features from a single, frozen pre-trained VFM—whether self-supervised or language-supervised—are sufficient for both image reconstruction and generation. Through a novel region-adaptive quantization that leverages these rich semantic features, VFMTok achieves several key advantages: it 1) accelerates AR model convergence, 2) enables high-fidelity CFG-free image synthesis, and 3) reduces the overall computational budget (Table 3 of the manuscript).

---

> > ### Author Response · Authors · 2025-08-09
> >
> > Dear Reviewer vMYn,
> >
> > Thank you again for your insightful and constructive feedback, which has been invaluable in strengthening our work. We have conducted new experiments and provided a comprehensive explanation in our rebuttal to address the issues you raised. As the discussion window closes, we would be grateful to know if your concerns have been fully addressed.
> >
> > Thank you once again for your time, effort, and thoughtful consideration.
> >
> > Sincerely,
> >
> > Authors of Paper 7970

---

### Official Review · Reviewer_SUoJ · 2025-06-30

**Clarity:** 3
**Significance:** 3
**Originality:** 3
**Rating:** 4
**Confidence:** 4

**Summary:**

This paper proposes VFMTok, an image tokenizer built on frozen vision foundation models. It introduces region-adaptive quantization and a semantic reconstruction objective to reduce redundancy and preserve semantics. The method improves image generation quality, achieves a gFID of 2.07 on ImageNet, and enables class-conditional synthesis without classifier-free guidance.

**Questions:**

See weaknesses.

**Ethical Concerns:**

["NO or VERY MINOR ethics concerns only"]

**Final Justification:**

I have carefully reviewed the authors’ rebuttal and found that it successfully addresses my concerns. While I find the use of VFM to improve quantization not particularly novel, the strong performance achieved without classifier-free guidance is quite surprising. I believe there may be an underlying reason behind this phenomenon that is worth deeper investigation. Therefore, I am currently inclined to raise my score to ``Borderline Accept''. I encourage the authors to further analyze and explore this aspect in future work.

**Limitations:**

The authors discussed both the limitations and potential negative social impacts.

**Quality:**

3

**Strengths And Weaknesses:**

**Strengths:**

- The paper is well-written and easy to follow
- The method achieves competitive reconstruction quality and generation quality.
- The paper achieves strong generative performance without relying on classifier-free guidance.



**Weaknesses:**

- The paper lacks novelty, as using vision foundation models as image tokenizers to improve VQGAN has already been explored in prior work[1][2].
- Could you provide a detailed explanation of why the model achieves better performance without classifier-free guidance than those that incorporate it?
- The paper lacks a thorough analysis of the *Region-adaptive Token Generation* module. While the authors claim that it “adaptively samples regions of similar patterns and extracts their VFM features for quantization,” it remains unclear whether the model truly behaves in this way. A deeper investigation into which aspects of the multi-level features are most beneficial for reconstruction would be important to support this claim and better understand the effectiveness of the approach.
- According to TiTok, MaskGIT-style generation offers higher efficiency compared to autoregressive methods. Has the author considered or experimented with such a decoding strategy for this work?



----

[1] Chen Z, Wang C, Chen X, et al. Semhitok: A unified image tokenizer via semantic-guided hierarchical codebook for multimodal understanding and generation[J]. arXiv preprint arXiv:2503.06764, 2025.

[2] Qu L, Zhang H, Liu Y, et al. Unified image tokenizer for multimodal understanding and generation[J]. arXiv preprint arXiv:2412.03069, 2024.

---

> ### Author Rebuttal · Authors · 2025-07-31
>
> Dear reviewer `SUoJ`,
>
> Thank you for reviewing our paper. Your concerns are answered as follows.
>
> > **W1:** The paper lacks novelty, as using VFMs as image tokenizers to improve VQGAN has already been explored in prior work (SemHiTok and TokenFlow).
>
> - Our motivation is to show that **frozen** VFM features **alone** are sufficient to build an image tokenizer.
> - Instead, both SemHiTok and TokenFlow adopt a dual-encoder approach, having one VQGAN branch for pixel-level features and a VFM branch for high-level semantics. VFMTok leverages the multi-level features from a single-branch VFM to capture both low-level and high-level features that benefit image reconstruction and generation.
> - Moreover, we also designed a light-weight region-adaptive quantization strategy to reduce redundancy in irregular image regions.
> - Besides, we provide a more detailed discussion on the primary purpose of the VFM in our response to Reviewer vMYn's first question (`vMYn`-**W1**). We kindly ask you to refer to that for more information.
> - Together, VFMTok sets apart prior works in both motivation and model architecture.
> - Thank you for bringing these works to our attention, and we will add a citation for discussion.
>
> > **W2:** Why does the model achieve better performance without classifier-free guidance than those that incorporate it?
>
> - Our intuition is that introducing semantic information renders the latent tokens semantically aware and helps stabilize the sampling process of the autoregressive (AR) model, thereby reducing the reliance on classifier-free guidance (CFG).
> - As shown in our results, the performance in both settings is comparable, with the CFG-free setup achieving slightly better gFID. However, we also observe that incorporating CFG slightly improves the gIS, indicating enhanced diversity. This reflects the typical trade-off between fidelity (gFID) and diversity (gIS) commonly seen in CFG-based approaches.
> - We will clarify these observations and the underlying trade-offs in the paper.
>
> > **W3.1:** Whether the region-adaptive token generation module adaptively sample regions of similar patterns?
>
> - Within the region-adaptive generation module, each region contains multiple sampling points. We utilize these points to sample from the last spatial features of the original DINOv2, creating a feature representation for each point.
> - These features are then aggregated via average pooling to form the representation for the entire region. The average similarity between each intra-region sampling point and the region feature is then computed. Additionally, we randomly sample points from outside the region, extract their features, and compute their similarity to the same region feature.
> - Below, $S_{in}$ and $S_{in}$ denote the sampling points within and outside a sampling region, respectively.
>
> | |  similarity($\uparrow$)|
> |:---:|:--:|
> | $S_{in}$| 0.860 |
> |$S_{out}$| 0.337 |
>
> As presented in the table above, it indicates a significant similarity among points *within* the region, whereas points from *outside* the region exhibit comparatively low similarity, which consolidates our hypothesis.
>
> > **W3.2:** Which aspects of the multi-level features are most beneficial for reconstruction?
>
> According to the suggested setup, we begin by conducting an ablation study to evaluate the impact of each layer on image reconstruction and generation. Subsequently, we cumulatively add each feature layer to observe the combined effect.
>
> The results, as presented in the table, show that while a single feature layer offers no significant advantage, the quality of both image reconstruction and generation markedly improves as more layers are incorporated.
>
> All tokenizers and AR generation models (VFMTok-L) are trained for 50 and 100 epochs, respectively. As shown in the tables, our results show that while individual layers provide little benefit, cumulatively adding more layers significantly improves both image reconstruction and generation quality. $F_i$ represents the indexed feature level.
>
> (a) Single-level feature investigation
> | $F_i$| rFID($\downarrow$)|rIS($\uparrow$)|gFID($\downarrow$)|gIS($\uparrow$)|
> | :---:|:---:|:---:|:---:|:---:|
> |   1   | 1.04 | 186.4 | 3.84 | 257.9 |
> | 2|  0.95 | 200.6 | 3.79 | 272.7 |
> |3 | 1.03 |  208.5 | 3.69 | 274.9 |
> | 4|  1.23 | 214.8 | 3.64 |  277.7 |
>
> (b) Analysis of cumulatively adding multi-level features
> | $F_i$| rFID($\downarrow$) |rIS($\uparrow$)|gFID($\downarrow$)|gIS($\uparrow$)|
> | :---:|:---:|:---:|:---:|:---:|
> |   1   | 1.04 | 186.4 | 3.84 | 257.9 |
> |   +2  | 0.94 | 205.0 | 3.69 | 274.4 |
> |   +3  | 0.94 | 210.8 | 3.27 | 272.5 |
> |   +4  | 0.89 | 215.4 | 3.09 | 274.2 |
>
> > **W4:** Has the author considered or experimented with MaskGIT decoding strategy for this work?
>
> We added an experiment that integrates VFMTok into MaskGiT [4]. For fairness, the images are generated without Classifier-Free Guidance (CFG).
> | Approach  | gFID($\downarrow$)| gIS($\uparrow$)|
> | :---|:---:|:---:|
> |MaskGiT [4] | 6.18 | 182.1|
> |MaskGiT [4] + VQGAN[43]| 7.25 |139.8|
> |MaskGiT [4] + VFMTok|**4.87**|**194.0**|

---

> > ### Comment · Reviewer_SUoJ · 2025-08-03
> > **Official Comment By Reviewer SUoJ**
> >
> > Thank you for the rebuttal. Most of my concerns have been resolved.  After reading your response to Reviewer vMYn,  I still find the novelty of the work to be somewhat limited.

---

> ### Author Response · Authors · 2025-08-04
>
> Could you kindly clarify which specific aspects of our approach or the related works led to the observation that VFMTok's novelty may be somewhat limited? It would enable us to better address your concerns more effectively in our response.
>
> 1. Specifically, the novelty of VFMTok can be summarized below:
> - VFMTok is the first to throughly validated that features from mainstream pre-trained VFMs **alone**, spanning both self-supervised and language-supervised paradigms, are effective for image reconstruction and generation.
> - To address the inherent redundancies in VFM's features, VFMTok introduces a novel region-adaptive quantization to achieve compact tokenization. This design substantially reduces the number of visual tokens while simultaneously enhancing performance, facilitating efficient AR generation without sacrificing quality (Tab. 1 in the manuscript).
> - Leveraging the rich semantics of **frozen pre-trained VFM** features and novel quantization techique, VFMTok enables AR models to converge faster and achieve high-fidelity, **CFG-free** image synthesis without complex heuristic designs (Tab. 3 in the manuscript).
>
> 2. Empirically, none of prior works showed simply using VFM features can achieve superior image reconstruction and generation performance. A key example is SemHiTok (similar to TokenFlow in motivation, methodology), which finds that VFM tokens lack pixel-level detail and thus requires a separate pixel branch to compensate for this deficiency. VFMTok, in contrast, circumvents this issue, achieving excellent image reconstruction and generation quality.

---

> > ### Comment · Reviewer_SUoJ · 2025-08-04
> > **Official Comment By by Reviewer SUoJ**
> >
> > Thank you for the clarification. I am happy to raise my score accordingly. I find it particularly interesting that the model performs well without classifier guidance—this is quite surprising. I encourage the authors to further explore and highlight this aspect in the paper.

---

> > > ### Author Response · Authors · 2025-08-04
> > >
> > > We are grateful for your kindness. We'll further explore and highlight this direction of AR generation without CFG in the final version.

---

### Official Review · Reviewer_APhE · 2025-06-30

**Clarity:** 3
**Significance:** 2
**Originality:** 2
**Rating:** 3
**Confidence:** 4

**Summary:**

This paper proposes VFMTok, an image tokenizer built on frozen vision foundation models like DINOv2 and CLIP. It introduces region-adaptive quantization to reduce feature redundancy and a semantic reconstruction objective to preserve high-level semantics. VFMTok enables efficient autoregressive image generation, achieving 3× faster convergence over baselines.

**Questions:**

Please refer to the weakness.

**Ethical Concerns:**

["NO or VERY MINOR ethics concerns only"]

**Final Justification:**

After reading the authors' rebuttal and the responses from other reviewers, I have decided to lower my score to borderline reject. Using VFM as a tokenizer is not uncommon, and while the method introduces high-level semantics, it provides limited evaluation on semantic understanding tasks, with only moderate results on image classification. On the generative side, under a fair setting with 256 tokens, the performance lags significantly behind state-of-the-art autoregressive models (as shown in Appendix Table A1).

**Limitations:**

yes

**Quality:**

3

**Strengths And Weaknesses:**

Strength:
1. This paper is easy to follow.
2. The authors introduce deformable sampling to improve token representation efficiency by enabling flexible, content-aware feature selection. This approach reduces redundancy in dense encoder features and naturally incorporates multi-scale information, enhancing the expressiveness of the resulting tokens.
3. The authors conduct ablation studies on various vision foundation models and core components of their framework, validating the effectiveness of each design choice.

Weakness:
1. The use of vision foundation model (e.g., DINO v2, CLIP, SigLIP) pretraining for tokenizers is not novel, as several recent works—particularly in the VAE/VQVAE domain (e.g., LightningDiT, VILA-U, VQGAN-LC, SoftVQ-VAE)—have explored similar directions.
2. The generation results are not particularly impressive, showing only limited improvements over the baseline LlamaGen. Moreover, the paper lacks comparisons with more recent methods, such as PAR and RandAR from CVPR 2025. Notably, the proposed tokenizer does not outperform non-VFM-based tokenizers and in some cases even underperforms them. Additionally, linear probing on ImageNet-1K using tokens produced by the VFM-based tokenizer yields significantly lower accuracy compared to directly using the original VFM features. These observations raise concerns about the overall effectiveness of the approach.
3. Although deformable sampling reduces FLOPs theoretically, its dynamic offset and weight calculations can cause slower inference in practice due to complex computations, limited parallelization, and inefficient memory access. The paper lacks an experimental comparison of inference speed to assess this issue.
4. There is a concern regarding the evaluation setting: generating images at 384×384 resolution and then resizing to 256×256 for evaluation may lead to unfair comparisons, potentially inflating performance metrics.

---

> ### Author Rebuttal · Authors · 2025-07-31
>
> Dear reviewer `APhE`,
>
> Thank you for your time and efforts in helping us improve our paper! We hereby answer your questions and resolve your concerns as follows:
>
> > **W1:** The use of VFM for tokenizers is not novel. LightningDiT, VILA-U, VQGAN-LC, SoftVQ-VAE have explored similar directions.
>
> Thank you for bringing these works to our attention. Despite using VFM for VAE/VQVAEs, we note clear distinctions from these works. We would like to clarify that our motivation is to show that **frozen** pre-trained VFM features **alone** are sufficient to build an image tokenizer. To achieve this, we leverage multi-level features from the VFM, and reduce redundancy via the region-adaptive quantization strategy. In contrast:
>
> - Approaches like LightningDiT and SoftVQ-VAE use a pre-trained VFM only as a feature distillation target for optimizing its tokenizer.
> - VQGAN-LC investigates initializing its codebook by heuristically clustering CLIP patch features, and its reconstruction and generation quality is substantially lower than VFMTok's.
> - VILA-U initializes its tokenizer's encoder with a pre-trained VFM, whose parameters are explicitly updated during training.
>
> In conclusion, all of them cannot build a tokenizer with frozen VFMs solely. We are sorry for the confusion will incorporate the discussion into the manuscript for a clearer relation with related works. Besides, we provide a more detailed discussion on the primary purpose of the VFM in our response to Reviewer vMYn's first question (`vMYn`-**W1**). We kindly ask you to refer to that for more information.
>
> > **W2.1:** Lack of comparison with PAR and RandAR.
>
> Per your request, we integrated VFMTok into RandAR and also compared it with that of PAR. As shown in the table below, VFMTok achieves better image reconstruction and generation performance. We will incorporate the comparison to the manuscript for a more complete comparison with recent works.
>
> |Approach | #code| #tok |rFID($\downarrow$) | rIS($\uparrow$)|gFID($\downarrow$)|gIS($\uparrow$)|
> |      :---      |    :---:     |      :---:     |     :---:   |      :---:    |    :---:   |   :---:  |
> |RandAR+VQGAN[43]| 16384 |  256 | 2.19    | 172.4| 2.55 | 288.8 |
> |RandAR + MaskGiT[4] |1024 |256 |2.28 |171.4 |2.47 | 271.0|
> |PAR + VQGAN[43]|16384| 256 |2.19 | 172.4| 3.76| 218.9|
> |RandAR + VFMTok|16384|256 |**0.89**|**215.4**|**2.25**|**300.6**|
>
> > **W2.2:** VFMTok does not outperform non-VFM-based tokenizers; linear probing accuracy is lower than original VFM features.
>
> In Table 2 of the manuscript, the image reconstruction quality of VFMTok is inferior to that of ImageFolder [26]. This is because the experiment setting is aligned with LlamaGen [43] baseline for a fair comparison. Once the codebook size and the number of tokens are aligned with those of ImageFolder [26], VFMToks's image reconstruction and linear probing accuracy surpass ImageFolder's, while its generation quality is comparable. The results aligned with ImageFolder are shown below.
>
> |Tokenizer |    #code.     |       #tok.     |    rFID($\downarrow$)    |       rIS($\uparrow$)  |  #epochs  |    gFID($\downarrow$)    |    gIS($\uparrow$)   | top-1($\uparrow$)|
> |    :---:     |    :---:     |      :---:     |    :---:   |      :---: |   :---:   |    :---:   |   :---:  |:---:  |
> |ImageFolder[26]| $2\times16384\times32$  |     $286\times2$     | 0.69 |  201.5 | 300 | 2.60 |295.0 |58.1|
> |VFMTok|   $32768\times12$   | 	  400	    | **0.60**  | **221.6** |    200    | **2.58** | 273.7 |**74.9** |
>
> Besides, there is a concern that VFMToks's linear probing accuracy is lower than that of the original VFM. This occurs because the quantization introduces information loss, consequently compromising the semantic fidelity of the tokens.
>
> > **W3:** Deformable sampling can cause slower inference; lacking experimental comparison of inference speed.
>
> - Please kindly note that deformable sampling is adopted only in the tokenizer's training stage. During inference (image generation), only the decoder of the tokenizer is used.
> - The question might be whether deformable sampling makes VFMTok slower to train. Thanks to the use of frozen VFM features, these can be cached, and the trainable module in the encoder is only a lightweight deformable sampler. Together, the training of our tokenizer is also substantially faster.
> - Below, we provide a complete comparison of the compute and throughput of different stages of the tokenizer across multiple resolutions. Both training and inference are efficient, especially as resolution increases.
>
> | Tokenizer     | Resolution | Encoding GMac | Decoding GMac | Training throughput ($\uparrow$, img/sec) | Inference throughput ($\uparrow$, img/sec)|
> | :-- | :---:|    :---:      |     :---:     |      :---:        |        :---:     |
> |VQGAN[43]|  256 |    69.36      |     126.63    |      205.46       |        370.38    |
> | VFMTok  |  256 |    9.39       |     104.24    |      286.43       |        343.87    |
> | VQGAN[43]|  384 |    156.62     |     285.67    |      89.84        |        162.24    |
> | VFMTok  |  336 |    9.39       |     176.08    |      183.67       |        205.72    |
>
> > **W4:** Generating images at 384×384 resolution and then resizing to 256×256 may lead to unfair comparisons.
>
> - This follows a common setting proposed in LlamaGen[43] and PAR, where comparisons are often conducted by generating images at $384\times384$ resolution and subsequently resizing them to $256\times256$ for evaluation. This paradigm requires 576 image tokens to represent an image.
> - Instead, VFMTok generates at a $336\times336$ resolution and resizes to $256\times256$ for evaluation, representing each image with just 256 tokens and thereby accelerating the AR generation speed.
> - For fairness, we also report the $256\times256$ image generation performance in Appendix Table A1. Please kindly take a look for reference.

---

> > ### Author Response · Authors · 2025-08-03
> >
> > Please let us know if you have any remaining questions or points for further discussion. We are happy to address any of your concerns.

---

> > ### Comment · Reviewer_APhE · 2025-08-05
> >
> > Thank you for your detailed rebuttal. However, after carefully considering your responses and those from the other reviewers, I still have significant concerns regarding the novelty and empirical strength of the proposed approach. Using VFM as a tokenizer is not particularly uncommon, and while your method aims to incorporate high-level semantics, the evaluation on semantic understanding tasks remains limited, with only moderate performance on image classification. On the generative side, under a fair comparison using 256 tokens, the results fall notably short of current state-of-the-art autoregressive models (as shown in Appendix Table A1). As a result, I have decided to lower my score to a borderline reject.

---

> ### Author Response · Authors · 2025-08-05
>
> 1. **Comparison of fairness**. VFMTok's novelty is in its tokenizer design, an approach orthogonal to advancements in AR generative frameworks. Thus, we used an identical generative framework (LlamaGen[43]) for all tokenizers for a fair comparison. Moreover, this decision ensure a fair comparison by strictly constraining the image token count to 256, maintaining the computational overhead consistent in the AR generation time. As shown in Table A1 in the appendix, the performance of RandAR in the rebuttal, and the table.(a) below, under the same computational budget, VFMTok improves the generation performance of LlamaGen, MaskGiT, RandAR, respectively.
>
> (a). Incoperating different tokenizer with MaskGiT for image generation with CFG-free.
> | Approach | #epochs | #tok. | gFID$\downarrow$ | gIS$\uparrow$|
> | :---     | :---:   | :---: | :---: | :---: |
> | MaskGiT + VQGAN | 300 | 256 | 6.18 | 182.1 |
> | MaskGiT[4]+VQGAN[43]|300|256|7.25|139.8|
> | MaskGiT[4]+VFMTok|150|256|4.87|194.0|
> | MaskGiT[4]+VFMTok|300|256|**3.95**|**214.8**|
>
> 2. **Comparison with SoTA AR generative frameworks:** For fairness, we also integrate VFMTok with advanced AR generative frameworks, RandAR and RAR, in terms of the image resolution of $256\times256$.  ~~Now we are working on the results of RandAR and RAR, and will report them later in a separate thread.~~ As depicted in the table below, VFMTok achieved **SOTA** results compared to the previous excellent works (including diffusion models and AR generative models) in both image generation with and without CFG.  Again, VFMTok primarily focused on tokenizer design for AR generation instead of AR model architectural advancements. Besides, experiments illustrated improvements on different generative frameworks under the same computational overhead. We would like to learn from the reviewer specific superior tokenizers or systems to compare with.
> | Approach | #Res| #tok. |params.| gFID($\downarrow$, w/ cfg) | gIS($\uparrow$, w/ cfg)|gFID($\downarrow$, w/o cfg) | gIS($\uparrow$, w/o cfg)|
> | :--- |:---:| :---: | :---: | :---: | :---: |:---:|:---:|
> |DiT-XL/2|${256\times 256}$|--|675M|2.27|278.2|9.60|--|
> |SiT-XL/2|${256\times 256}$|--|675M|2.06|270.3|8.30|--|
> |SiT-XL/2+REPA|${256\times 256}$|--|675M|1.80|284.0|5.90|--|
> |ImageFolder|${256\times 256}$|${286\times2}$|362M|2.60|295.0|--|--|
> |RandAR-L+VQGAN[43]|${256\times 256}$|256|343M|2.55|288.8|11.59|103.9|
> |**RandAR-L+VFMTok**|${256\times 256}$|256|343M|2.19|301.1|2.87|**216.1**|
> |RAR-L+VQGAN|${256\times 256}$|256|461M|1.70|299.5|6.72|129.2|
> |**RAR-L+VFMTok**|${256\times 256}$|256|461M|**1.47**|**316.8**|**2.14**|213.1|
>
>
>
> 3. **Evaluation on semantics:** We would like to clarify that the goal is to transform VFMs into good tokenizers (for generation), rather than understanding ability alone. For image generation, the semantics enable faster convergence and notably better CFG-free image synthesis. For understanding, in our rebuttal, we demonstrated a clear superiority over ImageFolder[26] (74.9\% vs. 58.1\% top-1 accuracy), the other one tokenizer that incorporates DINOv2 with advanced quantization technique. This consolidates our tokenizer preserves the semantics in VFMs better. We will be glad to hear from the reviewer for suggestions on further comparisons.
>
> 4. **Novelty of the method:** VFMTok is the first to thoroughly validate that features from a **single** mainstream pre-trained VFMs **alone**, spanning both self-supervised and language-supervised paradigms, are effective for image reconstruction and generation. Leveraging the rich semantics of frozen pre-trained VFM features and novel quantization technique, VFMTok enables AR models to converge faster and achieve high-fidelity, **CFG-free** image synthesis without complex heuristic designs (Tab. 3 in the manuscript).

---

> ### Author Response · Authors · 2025-08-08
>
> Following your suggestion, we have integrated VFMTok with current advanced AR generative frameworks following a practical setup. Both VFMTok and AR models are trained and evaluated with image size of ${256\times256}$. As presented in the above table, VFMTok achieved **SOTA** results compared to previous excellent works (both diffusion and AR generative models).

---

> > ### Author Response · Authors · 2025-08-09
> >
> > Dear Reviewer APhE,
> >
> > We sincerely thank you for your thorough and constructive feedback, which has been instrumental in improving our work. In response to your valuable comments, we have conducted additional experiments and submitted a detailed rebuttal to address your concerns. As the discussion window is closing, we would greatly appreciate it if you could let us know whether our responses have sufficiently resolved the issue you raised.
> >
> > Thank you once again for your time, effort, and thoughtful consideration.
> >
> > Sincerely,
> >
> > Authors of Paper 7970

---

### Official Review · Reviewer_KT8u · 2025-07-01

**Clarity:** 2
**Significance:** 2
**Originality:** 2
**Rating:** 4
**Confidence:** 4

**Summary:**

This paper proposes using pre-trained vision foundation models (VFMs) as image tokenizers. To improve the tokenizer's performance, the authors introduce a region-adaptive quantization framework and a semantic reconstruction objective. Experimental results show that the proposed tokenizer, VFMTok, achieves competitive reconstruction performance and can improve the generation quality of LlamaGen on the ImageNet dataset.

**Questions:**

- What is the performance of VFMTok on higher-resolution images (e.g., 512x512 or 1024x1024)? The current experiments seem limited to lower resolutions.

- The proposed region-adaptive quantization framework appears to be a general method. Is it independent of using a VFM-based tokenizer? How would this framework perform if applied to other existing tokenizers?

- Could you provide some direct qualitative comparisons? For example, showing the same image reconstructed or generated using VFMTok versus other baseline tokenizers would make it easier to see the practical differences.

**Ethical Concerns:**

["NO or VERY MINOR ethics concerns only"]

**Final Justification:**

Thanks the author for the detailed rebuttal.  I'm in general satisfied with the rebuttal with the comparison with more baselines and the efficiency comparisons and willing to increase the score. However I still feel the missing comparison with the SOTA models makes the paper weaker.

**Limitations:**

Yes

**Quality:**

2

**Strengths And Weaknesses:**

Strengths

- The motivation to use pre-trained vision foundation models as the basis for an image tokenizer is an interesting idea.

- Experimental results on ImageNet demonstrate that the proposed VFMTok can improve the generation performance of the LlamaGen model.

- The paper includes adequate ablation studies that help verify the effectiveness of each component within the proposed VFMTok.

Weaknesses

- Missing Baselines: The paper is missing comparisons to many important and relevant baseline tokenizers. For a thorough evaluation, VFMTok should be compared against methods like VQ-VAE, MAGVIT-v2, and MAETok, among others.

- Limited Compatibility Testing: The tokenizer is only tested within the LlamaGen framework. It is unclear how it would perform with other modern autoregressive visual generation models, such as EMU, Janus Pro, or MAR. This limited scope makes it difficult to assess the general applicability of the tokenizer.

- Lack of Efficiency Analysis: The paper does not include any comparison of latency or inference speed against baseline tokenizers. This is an important aspect for a component like a tokenizer.

- The tokenizer's ability to scale and generalize to more realistic, complex tasks is not demonstrated. Key evaluations, such as performance in text-to-image generation scenarios, are missing.

---

> ### Author Rebuttal · Authors · 2025-07-31
>
> Dear reviewer `KT8u`,
>
> Thank you for your thorough review of our paper and detailed suggestions for empirical results. We have made every effort to add the evaluations, which provided a more complete validation of our method, and will be incorporated to improve the paper. Your concerns and questions are answered as follows:
>
> > **W1:** Missing comparisons to tokenizers like VQ-VAE, MAGVIT-v2, and MAETok.
>
> Thanks for your reminder. We are sorry that a fair comparison with the discrete tokenizers VQ-VAE and MAGVIT-v2 is not possible due to their absence of officially released checkpoints. Nevertheless, we are able to compare with the strongest suggested baseline MAETok. As MAETok is a continuous VAE, we also convert VFMTok into a continuous one (denoted as VFMVAE), train and measure its image reconstruction quality on ImageNet. A comparison is listed below. We’ll incorporate this into the manuscript for a more detailed comparison.
>
> |Tokenizer| rFID($\downarrow$)|rIS($\uparrow$)|
> | :--- |:---:|:---:|
> |MAETok|0.48|–|
> |VFMVAE |**0.46**|**224.4**|
>
> > **W2:** The tokenizer is only tested within the LlamaGen framework, and its performance with other AR frameworks like EMU, Janus Pro, or MAR is unclear.
>
> Please note that 1) both EMU and MAR utilize continuous VAEs, while VFMTok is a discrete tokenizer. Besides, 2) EMU and Janus Pro are only evaluated on text-to-image generation tasks, while related works [23, 26, 44, 52, 55], including ours, mainly verify their effectiveness through class-to-image generation tasks. Thus, it is hard to perform an apple-to-apple comparison with other tokenizers under these AR frameworks.
>
> During the limited rebuttal period, we managed to integrate VFMTok into a recent AR framework, RandAR. As listed below, this integration allows VFMTok to achieve better image reconstruction and generation performance.
>
> |Tokenizer|\#code|\#tok|rFID($\downarrow$)|rIS($\uparrow$)|gFID($\downarrow$)|gIS($\uparrow$)|
> | :---| :---: |:---:|:---:|:---:|:---:|:---:|
> |RandAR|16384|256|2.19|172.4|2.55|288.8|
> |RandAR+VFMTok|16384|256|**0.89**|**215.4**|**2.25**|**300.6**|
>
> To better verify the compatibility of VFMTok, we also integrated it into the MaskGiT[4] framework. For fairness, the images are generated without Classifier-Free Guidance (CFG). VFMTok presents consistent improvement, further supporting its compatibility with AR and non-AR generation frameworks.
>
> |Generator  | gFID($\downarrow$) | gIS($\uparrow$) |
> |:---|:---:|:---:|
> |MaskGiT[4] | 6.18 | 182.1|
> |MaskGiT[4]+VQGAN[43]| 7.25 |139.8|
> |MaskGiT[4]+VFMTok|**4.93**|**190.4**|
>
> > **W3:** Lack of comparison of latency or inference speed against baseline tokenizers.
>
> Below, we show the decoding (inference) computation and throughput of tokenizers under different resolutions, where VFMTok is faster at higher resolutions thanks to the fixed-length tokens. Note that the comparison does not consider the time for AR generation of tokens. When considered, VFMTok requires a fixed length of 256 tokens, while VQGAN requires #token quadratic to the resolution, the inference efficiency would be more profound.
> |Tokenizer|Resolution|Decoding GMac|Inference Throughput($\uparrow$,imgs/s)|
> | :--| :---:|:---:|:---:|
> |VQGAN[43]|256|126.63|370.38|
> | VFMTok|256|104.24|343.87|
> | VQGAN[43]|384|285.67|162.24|
> | VFMTok|336|176.08|205.72|
>
> > **W4:** Performance in text-to-image generation scenarios is missing.
>
> Thanks for your suggestion. Due to the limited rebuttal period, we could only supplement with class-to-image generation experiments. Besides, highly related works such as ImageFolder [26], VAR [44], ViTVQGAN [52], TiTok [55], and RQ-VAE [23] also primarily validate their effectiveness through class-to-image generation tasks. We will continue to work towards the goal of text-to-image generation in the future, which, however, might be out of the scope of this paper.
>
> > **Q1:** What is the performance of VFMTok on higher-resolution images?
>
> Per your request, we added an experiment on image generation at $512\times512$ resolution. All AR models (VFMTok-L) are trained for 100 epochs and tested without CFG. As shown below, our method is consistently stronger on higher resolution settings.
> |Generator| rFID($\downarrow$) | rIS($\uparrow$)|gFID($\downarrow$,w/o cfg)|gIS($\uparrow$, w/o cfg)| gFID($\downarrow$, w/ cfg)| gIS($\uparrow$, w/ cfg)|
> | :-- | :---:|:---:|:---:|:---:|:---:|:---:|
> | BigGAN|–|–|8.43|232.5|--|--|
> |  ADM|–|–|23.24|58.06 |--|--|
> |MaskGiT[4]|–|–|7.32|156.0 |--|--|
> |VQGAN[43]|**0.49**|211.1|14.50|77.1|4.03|224.7|
> |VFMTok|0.65|**233.7**|**2.60**|**191.4**|**3.08**|**285.0**|
>
> > **Q2:** The region-adaptive quantization framework appears to be general. How would it perform if applied to other tokenizers?
>
> For simplicity, we adapted VQGAN[43] by replacing its PatchGAN with DinoGAN and inserting a region-adaptive module, constructing VQGAN-D and VQGAN-D-RA. The training setup is: tokenizer (50 epochs), VFMTok-L (100 epochs). As shown in the table below, it also improves image generation quality.
> |Tokenizer| rFID($\downarrow$) | rIS($\uparrow$)   | gFID($\downarrow$)  |  gIS($\uparrow$)   |
> | :--|:---:|:---:|:---:|:---:|
> | VQGAN-D|**0.80**|193.2|3.46|**230.9**|
> | VQGAN-D-RA|0.88|**198.3**|**3.39**|230.1|
>
> > **Q3:** Could you provide some direct qualitative comparisons?
>
> We are sorry that due to the rebuttal policy, we cannot include new figures at present. We will incorporate them into the paper in a future revision following your suggestions.

---

> > ### Comment · Reviewer_KT8u · 2025-08-06
> >
> > Thanks for the detailed rebuttal. However, I'm still having concerns on whether the proposed VFMTok will work in more practical setups, and how it compares to Janus-Pro or Emu3.

---

> ### Author Response · Authors · 2025-08-08
>
> 1. For fairness and following the practical setup, we integrate VFMTok with advanced AR generative frameworks, RandAR and RAR, in terms of the image resolution of ${256\times256}$. As depicted in the table below, VFMTok achieved SOTA results compared to the previous excellent works (including diffusion models and AR generative models) in both image generation with and without CFG. This reveals VFMTok is a good tokenizer for AR image generation.
>
> | Approach | #Res| #tok. |params.| gFID($\downarrow$, w/ cfg) | gIS($\uparrow$, w/ cfg)|gFID($\downarrow$, w/o cfg) | gIS($\uparrow$, w/o cfg)|
> | :--- |:---:| :---: | :---: | :---: | :---: |:---:|:---:|
> |DiT-XL/2|${256\times 256}$|--|675M|2.27|278.2|9.60|--|
> |SiT-XL/2|${256\times 256}$|--|675M|2.06|270.3|8.30|--|
> |SiT-XL/2+REPA|${256\times 256}$|--|675M|1.80|284.0|5.90|--|
> |ImageFolder|${256\times 256}$|${286\times2}$|362M|2.60|295.0|--|--|
> |RandAR-L+VQGAN[43]|${256\times 256}$|256|343M|2.55|288.8|11.59|103.9|
> |**RandAR-L+VFMTok**|${256\times 256}$|256|343M|2.19|301.1|2.87|**216.1**|
> |RAR-L+VQGAN|${256\times 256}$|256|461M|1.70|299.5|6.72|129.2|
> |**RAR-L+VFMTok**|${256\times 256}$|256|461M|**1.47**|**316.8**|**2.14**|213.1|
>
>
> 2. Our work focuses on studying VFM for visual generation. Extending it to unified models like Janus-Pro and Emu3 lies beyond the scope of the current study and will be pursued in future research. In addition, the short rebuttal period has limited the resources and time available for such an extension. Nevertheless, given the encouraging results obtained thus far, we believe this direction holds strong promise and will be explored in our future work.

---

> > ### Author Response · Authors · 2025-08-09
> >
> > Dear Reviewer KT8u,
> >
> > We sincerely thank you for the thorough and constructive feedback that has helped us improve our manuscript. In response to your valuable comments, we have performed additional experiments and submitted a comprehensive explanation to address your concerns. As the discussion period nears its conclusion, we would greatly appreciate it if you could let us know whether our responses have adequately addressed resolved the issue you raised.
> >
> > Thank you once again for your valuable time, effort, and thoughtful consideration.
> >
> > Sincerely,
> >
> > Authors of Paper 7970

---

### Note · Authors · 2025-08-13

Dear SACs, ACs, and Reviers,

We thank all reviewers for their valuable comments and feedback. We appreciate the recognition of the strengths of our work, VFMTok, summarized as follows:

- Solid claim, clear motivation, and sound methodology design(`KT8u`, `APhE`, `SUoJ`).
- Region-adaptive strategies in VFMTok effectively reduce redundancy in latent features while enhancing token expressiveness, thereby improving image reconstruction and generation quality(`KT8u`, `APhE`, `SUoJ`, `vMYn`).
- Comprehensive experimental results reveal the effectiveness of each design choice(`KT8u`, `SUoJ`, `APhE`)
- Superior image generation without CFG when integrated into the AR generative framework(`SUoJ`, `vMYn`)

Below, we outline our responses, which we believe have sufficiently addressed their concerns:

|Reviewer|Concerns|Our Response|Feedback|
|:---|:---|:---|:---:|
|`KT8u`|**A**. AR generation following a practical setup. **B**. Limited compatibility evaluation on unified models for text-to-image generation performance. |1. We integrated VFMTok into more advanced generative frameworks, showing [boosted performance gains](https://openreview.net/forum?id=PESrAH82Zh&noteId=kStmouYWj2) and [state-of-the-art results](https://openreview.net/forum?id=PESrAH82Zh&noteId=OH0c4TE53c). 2. VFMtok focuses on image generation, while unified understanding and generation fall beyond the scope of this study. Time and resource limitations have also constrained such an extension. |Without reply|
|`APhE`|**A**. Limited semantic capability. **B**. Generating images with a resolution of $256\times256$ should be comparable with advanced AR generative frameworks|1. We clarified that VFMTok focuses on transforming VFMs into strong tokenizers, and its benefits for generation, not understanding. Nevertheless, VFMTok outperforms prior works in semantic capability. 2. We provide [state-of-the-art results](https://openreview.net/forum?id=PESrAH82Zh&noteId=OH0c4TE53c) following the practical setup ($256\times256$ image generation).|Overall satisfied without further feedback|
|`vMYn`|**A**. Distinction from current works. **B**. What makes a VFM a good visual tokenizer?|1. We explained the [conceptual and empirical differences](https://openreview.net/forum?id=PESrAH82Zh&noteId=CuSp7q3hJ6), and highlighted [the novelty](https://openreview.net/forum?id=PESrAH82Zh&noteId=YsiADzr8Jz)  of VFMTok. 2. We fully analyze what makes a VFM for a good tokenizer.|Overall satisfied without further feedback|

---

### Decision · Program_Chairs · 2025-09-17

**Decision:**

Accept (poster)

**Comment:**

All reviewers found the paper was well written and its core idea of using pre-trained vision foundation models (VFMs) as tokenizers to be solid, well-motivated, and interesting.
The reviewers agreed that the experimental setup provided a strong validation of the method and its components, and were impressed by the generation performance without relying on classifier-free guidance.

There were concerns about the novelty of the method compared to the large body of work on pretrained tokenizers. There were also concerns about the scope of the experimental setup that focuses only on image generation rather then image understanding - a natural thing to test when relying on tokens that come from a pretrained VFM model. In addition, the reviewers requested more information and experiments.

In the discussion period, the authors provided the main additional experiments that were requested by the reviewers, and further emphasized the novelty of the approach as the first to demonstrate that features from a frozen VFM alone are sufficient for high-quality generation. They also emphasized that the scope of the paper was to test image generation only (without testing any semantic understanding tasks).

While most of the reviewers acknowledged that the main concerns were addressed and raised their score to recommend acceptance, one reviewer is still of the opinion that the novelty compared to prior work is limited, in particular because the semantic capacity of the tokens are not tested.

After reading all the reviews and discussions with the authors and among reviewers, I think this paper should be accepted, provided that the authors update the final version with all the results and clarifications discussed.